# A volumetric census of the Barents Sea in a changing climate

Sylvain Watelet[1,4], Øystein Skagseth[2], Vidar S. Lien[2], Helge Sagen[2], Øivind Østensen[2], Viktor Ivshin[3],
and Jean-Marie Beckers[4]

[1]Observation Scientific Service, Royal Meteorological Institute, Brussels, Belgium [current affiliation]
[2]Institute of Marine Research, Bergen, Norway
[3]Polar Branch of Russian Federal Research Institute of Fisheries and Oceanography (PINRO), Murmansk, Russia
[4]Department of Astrophysics, Geophysics and Oceanography, GeoHydrodynamics and Environment Research Unit, FOCUS
Research Unit, University of Liège, Liège, Belgium

**Correspondence:** Sylvain Watelet (swatelet@uliege.be)

**Abstract.**

The Barents Sea, located between the Norwegian Sea and the Arctic Ocean, is one of the main pathways of the Atlantic
Meridional Overturning Circulation. Changes in the water mass transformations in the Barents Sea potentially affect the ther-
mohaline circulation through the alteration of the dense water formation process. In order to investigate such changes, we
present here a seasonal atlas of the Barents Sea including both temperature and salinity for the period 1965–2016. The atlas is
built as a compilation of datasets from the World Ocean Database, the Polar Branch of Russian Federal Research Institute of
Fisheries and Oceanography, and the Norwegian Polar Institute using the Data-Interpolating Variational Analysis (DIVA) tool.
DIVA allows for a minimization of the expected error with respect to the true field. The atlas is used to provide a volumetric
analysis of water mass characteristics and an estimation of the ocean heat and freshwater contents. The results show a recent
"Atlantification" of the Barents Sea, that is a general increase of both temperature and salinity, while its density remains stable.
The atlas is made freely accessible as user-friendly NetCDF files to encourage further research in the Barents Sea physics
(https://doi.org/10.21335/NMDC-2058021735, Watelet et al. (2020)).

## 1 Introduction

The Barents Sea shelf is a "hotspot" in the ongoing, rapid climatic changes taking place in the Arctic (Lind et al., 2018). During
recent decades, the Barents Sea has (BS) contributed most of the reduction in Arctic winter sea-ice cover (Yang et al., 2016).
Moreover, the northern, Arctic–dominated part of the Barents Sea has experienced an "Atlantification" (or "borealization") with
profound impact on its physical conditions, such as water mass transformations and properties (Lind et al., 2018), as well as on
biology and marine ecosystem (Fossheim et al., 2015). As the northern limb of the Atlantic Meridional Overturning Circulation
(AMOC) and a source for dense Arctic Intermediate Water (Schauer et al., 1997), changes to the water mass transformation

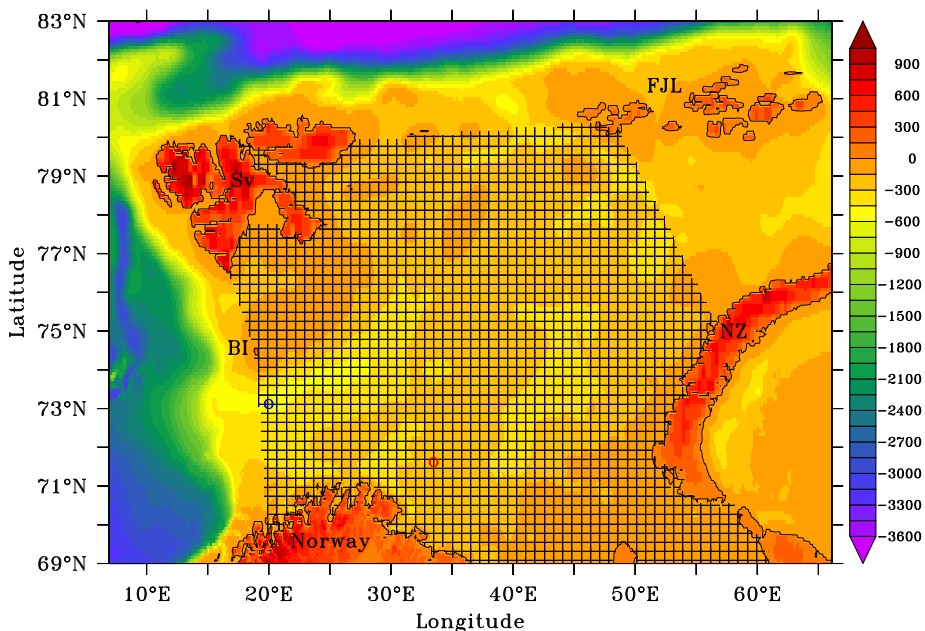

**Figure 1.** Bathymetry of the Barents Sea and its neighbouring seas. Our analyses on the Barents Sea correspond to the shaded region. The Barents Sea Opening, located between the Norwegian coast and Bear Island, and the Kola sections are shown as blue and red circles respectively. BI stands for Bear Island, Sv for Svalbard, FJL for Franz Jozef Land and NZ for Novaja Zemlja.

processes in the Barents Sea affect the thermohaline circulation of the North Atlantic and Arctic oceans (Swift et al., 1983; Kuhlbrodt et al., 2009; Mauritzen et al., 2013; Lozier et al., 2019).

The Barents Sea is the largest shelf sea of the Arctic Ocean, and it is bounded by Norway and the Kola Peninsula (Russia) to the south, the Svalbard and Franz Josef Land archipelagos to the north, and Novaya Zemlya to the east (see Fig. 1). The

Barents Sea is connected to the Norwegian Sea to the west through the Barents Sea Opening (BSO), and to the Arctic Ocean to the north and northeast. Together with the Fram Strait between Svalbard and Greenland, the BSO is the main gateway between the North Atlantic and the Arctic and, thus, a main pathway for Atlantic Water transport northwards from the Nordic Seas to the Arctic Ocean (Knipowitsch, 1905; Helland-Hansen and Nansen, 1909). Due to its climatic importance and vast marine resources, the Barents Sea area is sampled and monitored on a seasonal timescale (Eriksen et al., 2018). However, the

coverage varies between seasons and years, especially during winter and spring, and the spatial coverage is sometimes only semi-synoptic or concentrated at fixed sections.

Satellite remote sensing provides observations of sea surface temperature, and recently sea surface salinity, with high resolution in both space and time. For example, using AVHRR data, Comiso and Hall (2014) found the northern Barents Sea to be one of the areas within the Arctic that shows the highest temperature increase for the period 1981–2012. Furthermore, they found

a significant decline in sea-ice cover between the two periods: 1979–1995 and 1996–2012. However, to investigate regional climate processes, such as water mass transformation and property changes, *in situ* observations are needed. *In situ* data often

have disadvantages of a limited coverage in space (e.g. repeated hydrographic sections) and/or time (e.g. ship surveys). Thus, providing these observations on a regular grid is desirable in order to examine spatio-temporal changes.

Here, we present a gridded dataset of temperature and salinity in the Barents Sea region at seasonal temporal resolution for the period 1965–2016, based on all available *in situ* observations. The dataset is compiled using the Data-Interpolating Variational Analysis (DIVA) tool. We provide the dataset including fields of expected error, and present two examples of usage where this gridded dataset has an advantage over the non–gridded raw data: volumetric analysis of water mass characteristics, and estimation of ocean heat and freshwater content.

## 2 Data sources

*In situ* hydrographic data were obtained from three different sources, the World Ocean Database 2013 (WOD13), the Norwegian Polar Institute, and the Polar Branch of Russian Federal Research Institute of Fisheries and Oceanography (PINRO). The data consist mostly of Conductivity-Temperature-Depth (CTD) cast profiles, while data from the pre-CTD era (ca. mid–1970s) consist of Salinity-Temperature-Depth (STD) cast profiles as well as discrete samples. Expendable bathythermograph (XBT) data are also included. Data from CTD are usually provided at a vertical resolution of 1 meter, while some profiles are provided at a vertical resolution of 5 meters. Discrete samples are provided at standard depths where the vertical resolution varies with depth and increases from 5 meters near the surface to 50 meters near the bottom depth in the Barents Sea (around 200-300 m).

The hydrographic data obtained from WOD13 included data until 2016 and were limited to the area 7°E–66°E, 68°N–83°N. Only data with a quality control flag value of 0 (i.e., accepted cast) were included.

Hydrographic data from the Norwegian Polar Institute, which are not included in the WOD13 database, include CTD casts from 1998, 2003, 2004, 2005, and 2011. These data only included post-processed, quality-controlled data with a quality flag value of 1 ("good data").

From the hydrographic data obtained from PINRO, which cover the period 1965-2014, only data with a quality control flag value of 1 ("good data") were included. These data complement CTD data from the Institute of Marine Research already available from the WOD13 with respect to geographical coverage from joint surveys in winter and summer.

The data coverage is usually better in the spring (Feb-Mar-Apr) and autumn (Aug-Sep-Oct) seasons compared with the rest of the year due to extensive survey activity during these seasons. However, while the surveys generally cover the ice-free area of the Barents Sea, the spatial coverage vary between years and the coverage is usually more extensive in the autumn compared with the spring. Moreover, while data from the annual spring and autumn surveys in the Barents Sea are obtained on a regular grid, data from other surveys are more focused in smaller areas or along fixed sections.

## 3 Software and method

Ocean Data View (ODV) software was used to convert the hydrographic data files into a format readable by the DIVA software, the ODV spreadsheet (https://www.bodc.ac.uk/resources/delivery_formats/odv_format/).

DIVA is a statistical software designed to generate continuous fields from heterogeneously distributed *in situ* data using a Variational Inverse Method (Brasseur, 1995; Troupin et al., 2012). The result of its variational analysis are gridded fields which minimise the expected errors with respect to the unknown true fields. Under a few assumptions on the correlations, the Variational Inverse Method (VIM) is equivalent to the popular Optimal Interpolation (Rixen et al., 2000). In practice, the aim of the VIM is to minimize the following cost function $J$:

$$J[\varphi] = \sum_{j=1}^{N_d} \mu_j [d_j - \varphi(x_j, y_j)]^2 + ||\varphi||^2$$

where the $N_d$ observations $d_j$ are used to reconstruct the analysed field $\varphi$ and with

$$||\varphi||^2 = \int_D (\alpha_2 \nabla \nabla \varphi : \nabla \nabla \varphi + \alpha_1 \nabla \varphi . \nabla \varphi + \alpha_0 \varphi^2) dD$$

where $\alpha_0$ penalizes the field itself (anomalies with respect to a reference field, e.g., a climatological average), $\alpha_1$ penalizes gradients (no trends), $\alpha_2$ penalizes variability (regularization), and $\mu_j$ penalizes data-analysis misfits (objective analysis) (Troupin et al., 2016).

Unless specified otherwise, we always use the command line version of DIVA in this study. This version comes with the full set of options, for instance regarding the optimization of the statistical parameters later used in the analyses.

Then, using DIVA preprocessing tools, the data were vertically interpolated onto 23 depths (500, 450, 400, 350, 300, 250, 200, 175, 150, 125, 100, 75, 50, 45, 40, 35, 30, 25, 20, 15, 10, 5, 0) following the Weighted Parabolas method (Reiniger and Ross, 1968). These levels were chosen in view of increasing the resolution next to the surface where the variability of both temperature and salinity are expected to be higher.

The Barents Sea bathymetry to be used in the atlas processing was extracted from the General Bathymetric Charts of the Oceans (GEBCO) at a spatial resolution of 30 seconds by using Diva-on-web (http://ec.oceanbrowser.net/emodnet/diva.html). This bathymetry was then smoothed to a resolution of 1/8° by using a 2D convolution low-pass filter followed by a linear interpolation to avoid too complex shapes when computing the coastlines for each depth level. Besides, several fjords were removed from the bathymetry. All the interpolated data falling outside these smoothed coastlines or outside the full domain (6.9–66.1°E ; 69–83°N) shown in Fig. 1 were removed. A data range check was also performed and excluded temperature data falling outside -1.9–20° C and salinity data outside 30–36. The remaining data availability per season is shown in Fig. 2 for temperature and in Fig. 3 for salinity.

For each of the 23 depth levels, the objective is to perform one analysis for each season and for each year between 1965–2016. Based on data availability from regular cruise activity, we chose the seasons as follows: November to January (winter), February to April (spring), May to July (summer) and August to October (autumn). The first season is thus November 1964 to January 1965, the last being August to October 2016. The analysis is carried out in two steps. A reference field, or a first guess state, needs to be created before each analysis is carried out. The reference fields are created by collecting all data for each season across 11 years centred around the year to be analysed. A moving window centred at the year of interest is used due

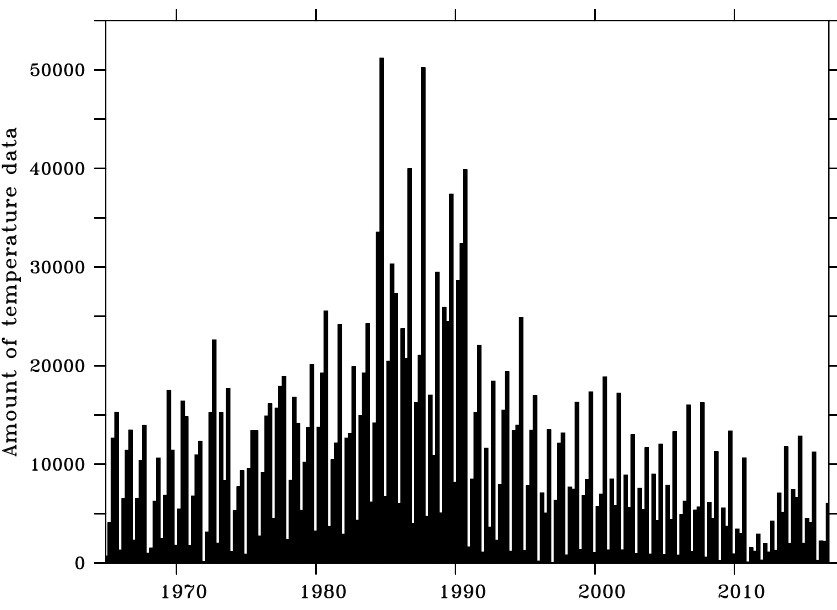

**Figure 2.** Availability of temperature data in the Barents Sea as a function of time (seasons).

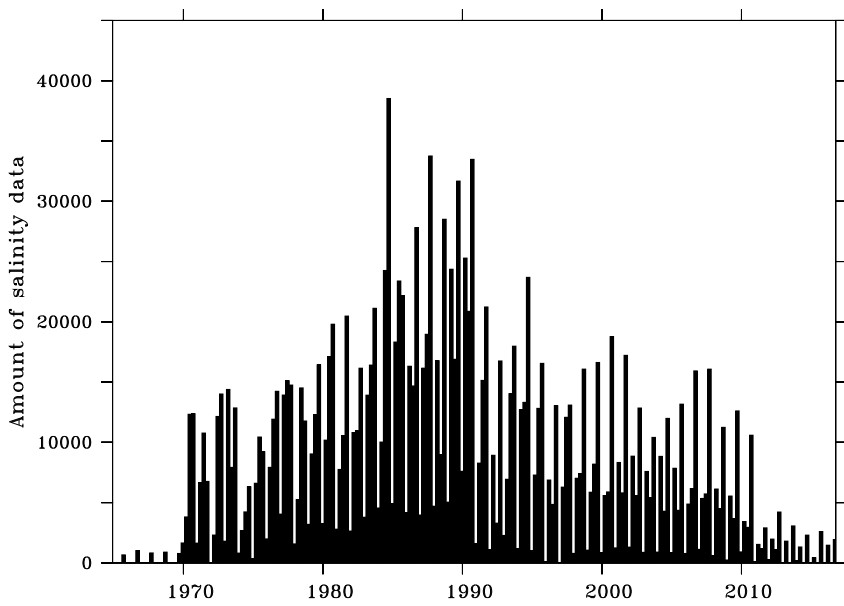

**Figure 3.** Availability of salinity data in the Barents Sea as a function of time (seasons).

to the strong multidecadal variability of the region (e.g. Smedsrud et al., 2013). Near the beginning and end of the period the window size is reduced to the available years (i.e., the reference field for 1965 is based on data from the period 1965–1970).

The horizontal average is used as a constant first guess when creating the reference fields. Therefore, 4 reference fields are generated per year, that is one per season. By subtracting the reference field from the original data, DIVA directly works with anomalies of temperature and salinity before adding back the reference to the optimal analysis. In this way, the analysis tends to smoothly reach the reference values in the absence of data.

In the reference fields, the correlation length is estimated by a fit between the empirical data correlation function as a function

of the distance and its theoretical counterpart, while the signal to noise ratio is approximated by cross validation techniques (Craven and Wahba, 1978). Both the correlation length and the signal-to-noise ratio are thus estimated on the basis of the data sets. Moreover, they are both filtered vertically to avoid unrealistic discontinuities between depth levels. To avoid an overconfidence in the data accuracy, the signal-to-noise ratio is capped at 10 for salinity and 3 for temperature, because of its higher temporal variability. Using these statistical parameters, the reference fields are computed by the Variational Inverse

Method with DIVA over the same 11 years, for each season.

Then, each analysis is performed using the corresponding 11 year-reference field and the associated statistical parameters. We decided to use the statistical parameters based on the larger amount of data (11 years) in order to increase their robustness and decrease their variability. For temperature, a logit transformation was applied on data beforehand, so as to ensure the results are constrained between -1.9 and 20°C after applying a reciprocal function to the analyses. This extra precaution for

temperature is justified by the sea ice formation around -1.9° C. The analyses are stored on an output grid with a resolution of 0.1° in latitude and 0.25° in longitude. Other atlas products, such as the WOA, are also provided on regular lat-lon grids, as well as most operational ocean models. Hence, it makes some of the usages more straightforward.

In order to assess the reliability of the analyses, an error field associated with each of them is computed by using the clever poor man's method, a good compromise between the computation time and the accuracy (see Beckers et al. (2014)). The poor

man's error is computed by analysing a "data" vector with unit values and is very cost-effective (Troupin et al., 2010), but the error field is too optimistic. It is shown that using the same method with a correlation length divided by a factor $\sim 1.7$ requires a similar computation time and yields a more realistic estimate of the error, that is, the clever poor man's error. This analysis error is then compared to the first guess error, and the ratio of those errors yields the relative error field which thus consists in a value between 0 and 1. Qualitatively, this figure measures the added value brought by *in situ* data to the analysis: 0 would be

the true field while 1 corresponds to an absence of data, that is an analysis equal to the first guess.

## 4   Temperature and salinity atlas

The temperature and salinity atlas is available at the Norwegian Marine Data Centre as two NetCDF files. Each file contains analyses of temperature or salinity, respectively, for all seasons and years at all depths, and also includes the error field associated with each analysis. The statistical parameters (correlation length and signal to noise ratio) and the analysed fields restricted

to the most reliable areas are also available. These latter analyses are masked if the relative error exceeds 0.3 or 0.5. As shown

in Fig. 2 and 3, there are several seasons with data gaps. In such cases, the atlas only contains a missing value, for both the analysis and the error field. The data gaps for salinity are mainly found before 1970 and after 2010, while the temperature has only exceptional data gaps. Between 1970–2010, there are data gaps in the salinity atlas during the 1971—1972 winter period and in both temperature and salinity atlas during the 1996–1997 winter period. Besides, other gaps appear sometimes in the deepest layers. In Section 5, we explain how to make use of the error field to take into account the data coverage before applying any analysis. The data is accessible at https://doi.org/10.21335/NMDC-2058021735 (Watelet et al., 2020).

The hydrographic atlas presented here complements global gridded data products, such as the World Ocean Atlas (Locarnini et al., 2018; Zweng et al., 2018), by providing a regional approach tailored to the specific region by offering a higher spatiotemporal resolution allowed by the higher regional data coverage. The presented gridded dataset provides researchers with readily available observation-based data, including error estimates, for several key purposes, such as numerical ocean model validation and regional climate studies. While point-based observations are useful for process studies and observation-model comparisons, a gridded dataset enables the researcher to easily conduct spatiotemporal analysis, such as empirical-orthogonal-function (EOF) analysis for a more robust measure of a numerical model's performance (e.g. Wang et al., 2014). Furthermore, a gridded dataset enables easy computation of integrated measures such as ocean heat content and ocean freshwater content (e.g. Lind et al., 2018), area covered by specific water masses (e.g. Johannesen et al., 2012), or overall changes in water mass characteristics (e.g. Skagseth et al., 2020) for regional climate studies.

## 5    Uncertainties and use of error field

In the following sections we demonstrate how the error field provided in the atlas can be utilized to objectively limit the data in time or space before applying the desired analysis. Moreover, we give some examples of possible usages of the atlas product.

### 5.1    Most reliable period

Lind et al. (2018) provided some evidence suggesting a warmer and saltier northern Barents Sea since the mid–2000s. Here, we show the changes in water mass characteristics in the whole Barents Sea based on the results from the atlas, by use of volumetric Temperature–Salinity (*T-S*) diagrams. We limit our analysis to comparing the two 5-year periods 1994–1998 and 2006–2010, where the former represents a relatively cold period while the latter represents a warm period relative to the last 50 years.

First, we consider uncertainties by investigating the error field from the atlas. As the data coverage in the Barents Sea varies between years, seasons and sub-regions, the error field varies accordingly (Fig. 4). The geographical patterns of the error fields are similar at other depths (not shown). Generally, the errors are larger in the northern and eastern parts of the Barents Sea compared with the western and southern parts, due to differences in data coverage (see Section 5.2; Fig. 4; Supplementary Material). Moreover, the data coverage is generally better in the autumn season and, hence, the error is generally smaller compared with the other seasons. For this reason, we decided to focus on the autumn only when considering the whole Barents

Sea. For studies needing the whole Barents Sea climatology in other seasons (e.g. winter), other data sources could prove necessary.

Volumetric *T-S* diagrams for both 1994–1998 and 2006–2010 were compiled by summing all the pixels falling inside the *T-S* classes defined by temperature ranging from -1 to 7 °C and salinity varying between 33 and 35.5, using steps of 0.05 °C and 0.025, respectively. In this calculation, each pixel is weighted by its vertical extent for each corresponding layer to get a proportional representation of to the water volume within each *T-S* class. Moreover, the horizontal extent of each pixel is weighted by the latitude $\varphi$ relative to the average latitude $\varphi_0$ of all the grid cells, due to the narrowing of the longitudinal bands towards the north, using the function

$$Weight = \frac{\cos \varphi}{\varphi_0}.$$

The average *T-S* properties in both periods is shown in Fig. 5a, while the difference between the two periods is shown in Fig. 5b. Clearly, both the temperature and the salinity increased, on average, from the 1990s to the 2000s in the whole Barents Sea, which is in line with the findings of, e.g., Skagseth et al. (2020). Between the T or S classes showing the highest change, there is temperature shift of 5° C and a salinity shift of 0.2. The density, however, remained more or less unchanged due to the cancelling effects of increasing haline contraction and thermal expansion on density, again consistent with the findings of Skagseth et al. (2020).

Further utilizing the error field, we provide an estimation of the uncertainties for both the two 5-year periods included in the above analysis. Comparing the error fields in both periods (Fig. 5c, d) with the changes in the *T-S* properties between the two periods (Fig. 5b), as well as the *T-S* diagrams of both periods (Fig. 5a), it is clear that the error is small for the *T-S* classes that have the largest presence and also are showing the largest changes. This strengthens the reliability of the findings of *T-S* changes in the Barents Sea in autumn.

## 5.2   Most reliable area

In this Section, we focus on the spatial pattern of the error field. We first limit our study area to the area where the average relative error for temperature is less than 0.5 (Fig. 6), hereafter referred to as the Most Reliable Area (MRA). Similarly to Section 5.1, salinity and temperature exhibit close error fields (not shown). We then average the relative error fields for all seasons (see Supplementary Material). Compared to the rest of the Barents Sea, the MRA shows relatively low uncertainties for all seasons due to the better data coverage. The MRA encompasses the southern part of the Barents Sea which is dominated by the Atlantic Water inflow and kept ice-free year round, hence the better data coverage in all seasons. This allows us to analyze all the seasons in the MRA, in contrast to only the autumn season when analyzing the whole Barents Sea (see section 5.1), with the exception that for salinity the data coverage is sufficient only for the period 1970—2010. For temperature, we use the period 1965–2015. In addition, there are gaps in the salinity data during the 1971—1972 winter period and in both temperature and salinity data during the 1996–1997 winter period.

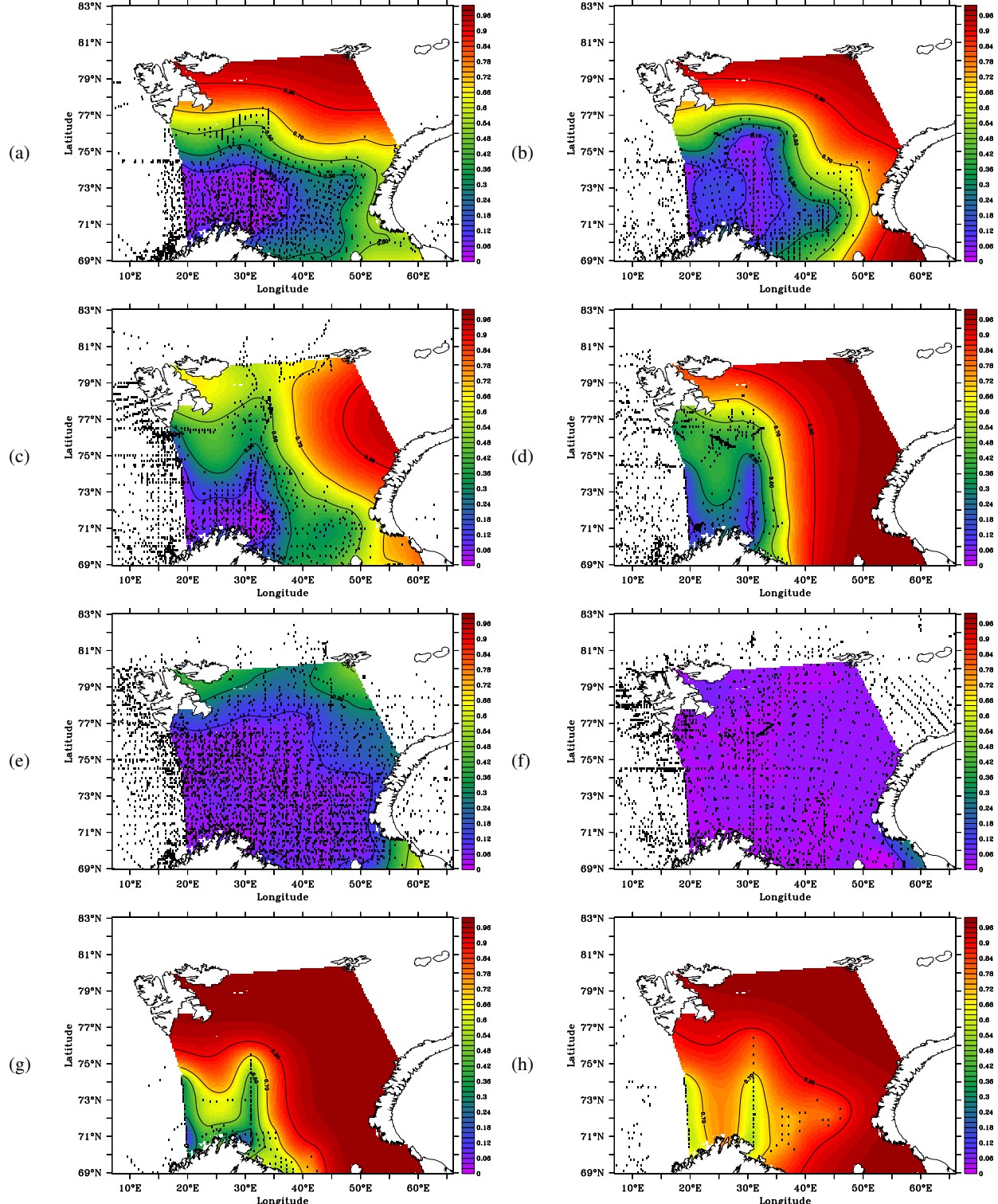

**Figure 4.** Average relative error for temperature at the Barents Sea surface between 1994–1998 (left column) and 2006–2010 (right column) between 1994–1998. (a) and (b) correspond to spring, (c) and (d) to summer, (e) and (f) to autumn, (g) and (h) to winter. This variable measures the added value brought by *in situ* data to the analysis: 0 would be the true field while 1 corresponds to an absence of data, that is an analysis equal to the first guess. The winter 1996–1997 was excluded from the computations due to a lack of data.

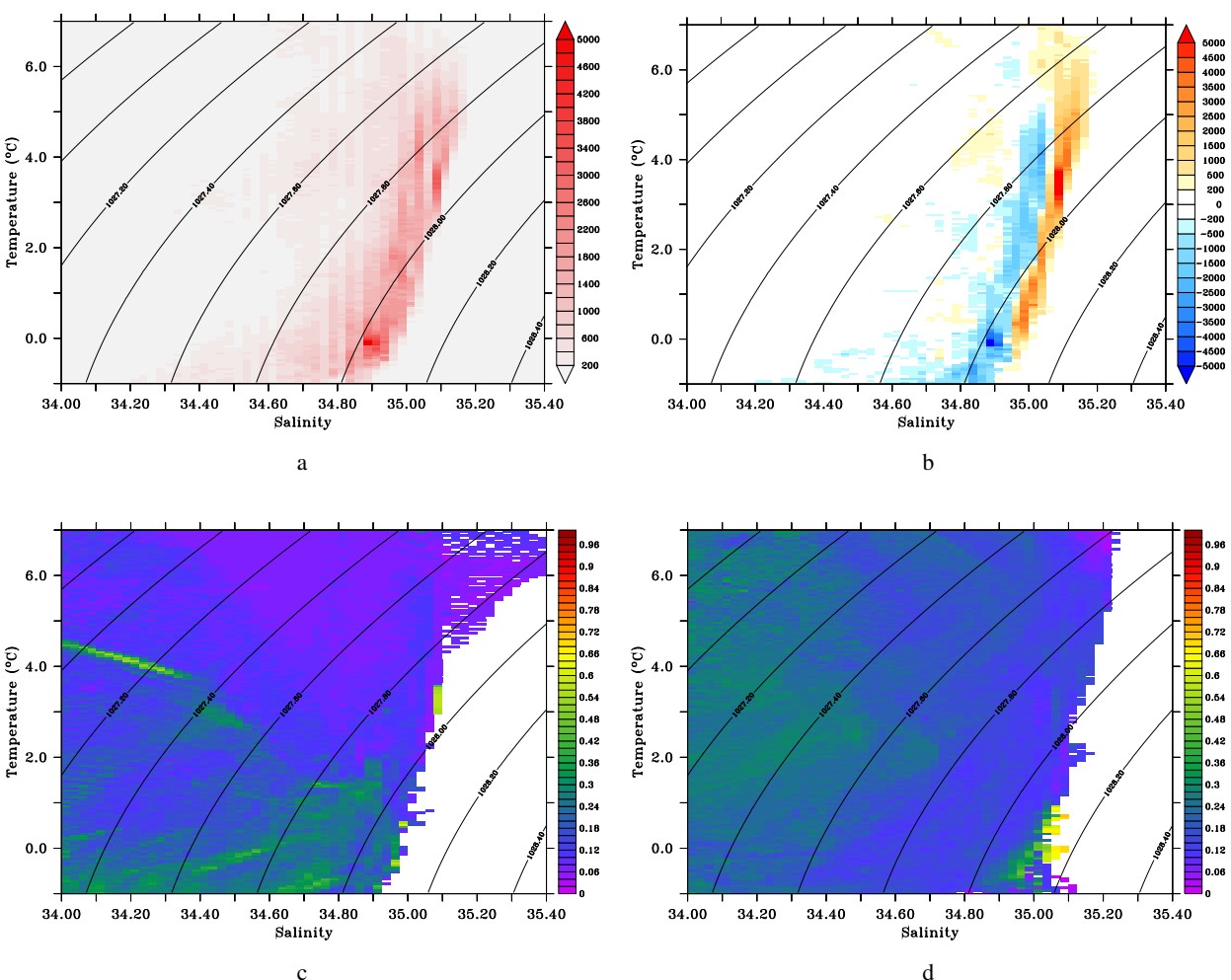

**Figure 5.** (a) Average of the volumetric *T-S* diagrams during both 1994–1998 and 2006–2010 periods. A value of 1 corresponds to a pixel with a vertical extent of 1 m at $\varphi_0 =$74.5°N. Isopycnals are shown for 0 m (black). (b) Difference in volumetric *T-S* diagrams between 2006–2010 and 1994–1998. (c) Average relative error weighted by the layer thickness and the latitude for each *T-S* class between 1994–1998. (d) Average relative error weighted by the layer thickness and the latitude for each *T-S* class between 2006–2010. For all panels, only autumn is used and the areas with errors above 0.99 were excluded from the computations to avoid contamination by small areas without data and disconnected from the sea.

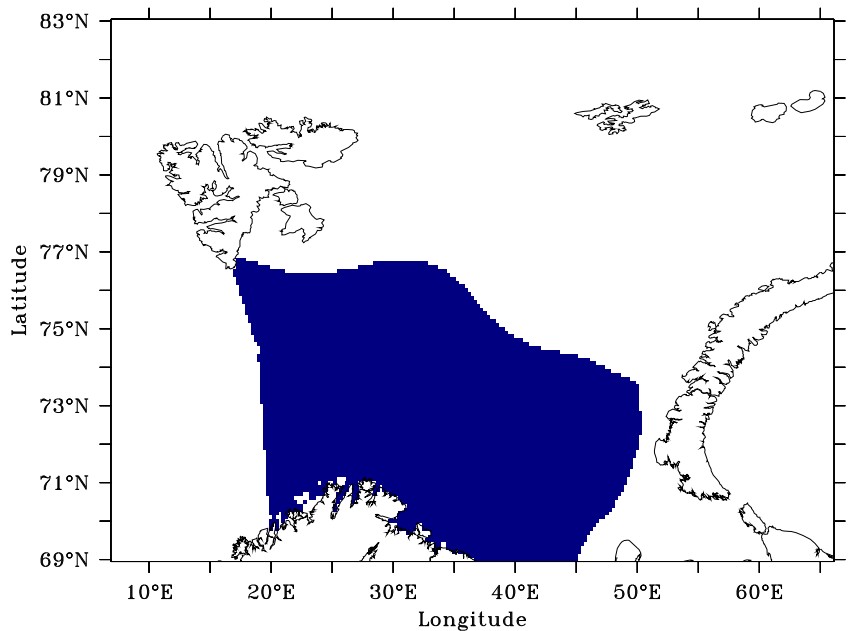

**Figure 6.** Most reliable area as defined from temperature and salinity relative errors.

We start the analysis of the MRA by investigating the water mass characteristics within the region represented by vertical profiles of temperature and salinity averaged over the MRA and for each season (Figs. 7, 8). The temperature gradually increased throughout the whole water column during the period 1965–2015, by 1.74° C on average. For salinity, matters are not so clear, except the unambiguous average increase of 0.11 between 1990–2010, similarly to the observation made for the whole Barents Sea between the 1990s and the 2000s. The potential density relative to the surface is shown in Fig. 9. There is no clear trend throughout the period, which indicates that the observed warming trend is compensated to some extent by a salinity increase. This result is consistent with the changes in the Barents Sea hydrographic properties reported by Skagseth et al. (2020) and also upstream in the Norwegian Sea (Mork et al., 2019).

Further analyses of volumetric changes in the MRA are performed in order to better assess the evolution of temperature, salinity and density classes throughout the water column. The calculations follow a method similar to Section 5.1 and are performed for each season between 1965–2015 for temperature and between 1970–2010 for both salinity and density. The aim is to show the relative volume occupied by each temperature and salinity class. Fig. 10 shows the evolution of temperature classes ranging from -1 to +7 °C with a step of 1 °C. There is a clear increase in the volume of the warmest temperature classes at the expense of the coldest classes throughout the period. For instance, between the periods 1975–1985 and 2005–2015, the relative volume occupied by temperature below 0° C decreased from 19.64% to 1.77%. Changes in salinity classes between 34.4 and 35.2 with a step of 0.1 are shown in Fig. 11. Here, matters are less clear but there is however an increase of salinity classes above 35 and a decrease of the lowest-salinity class between 1980–2010. For instance, between the periods 1975–1985 and 2000–2010, the relative volume occupied by salinity below 35 decreased from 86.84% to 62.67%. Moreover, the low salinity

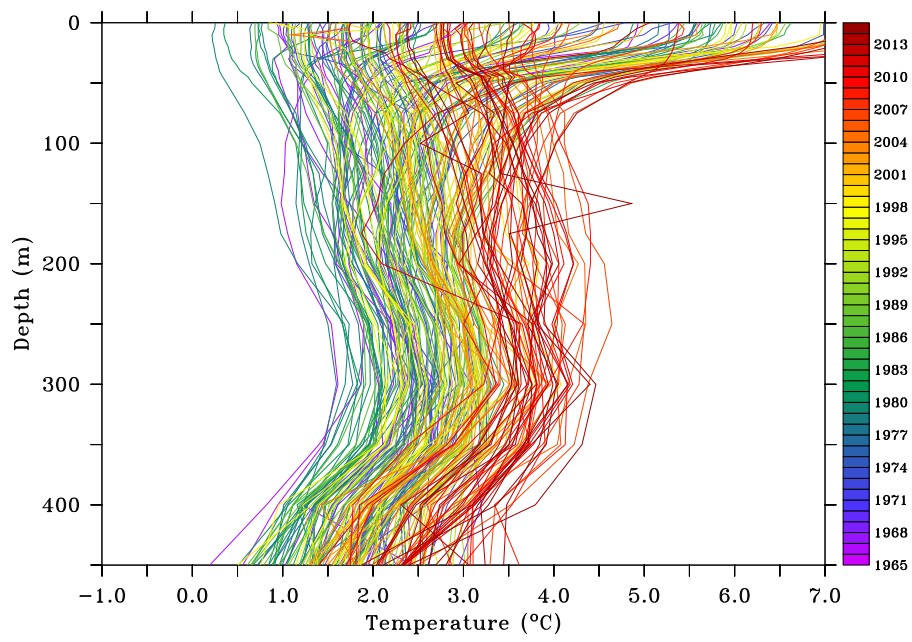

**Figure 7.** Seasonal averaged profiles of temperature on the most reliable area between 1965–2015.

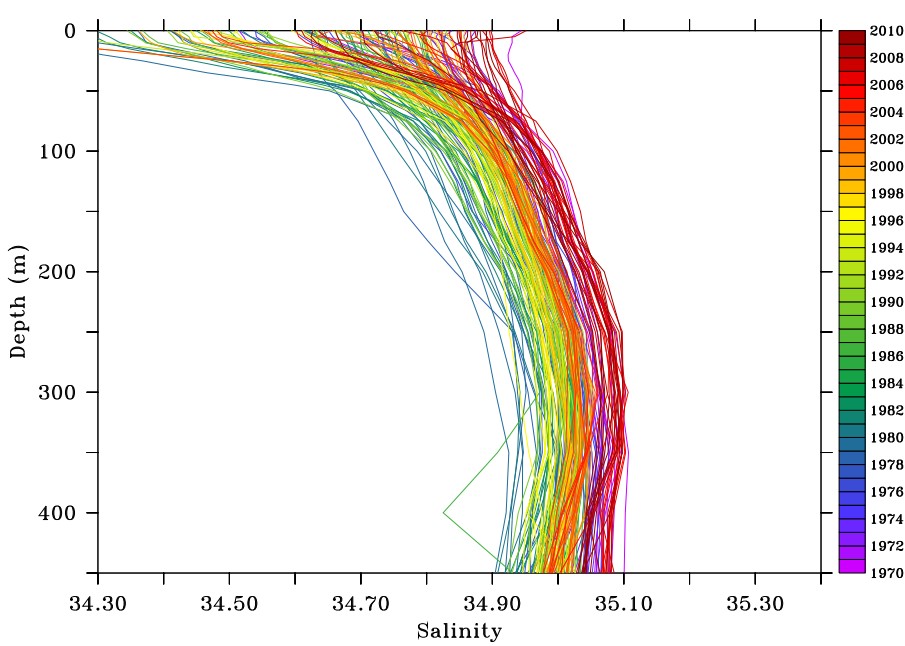

**Figure 8.** Seasonal averaged profiles of salinity on the most reliable area between 1970–2010.

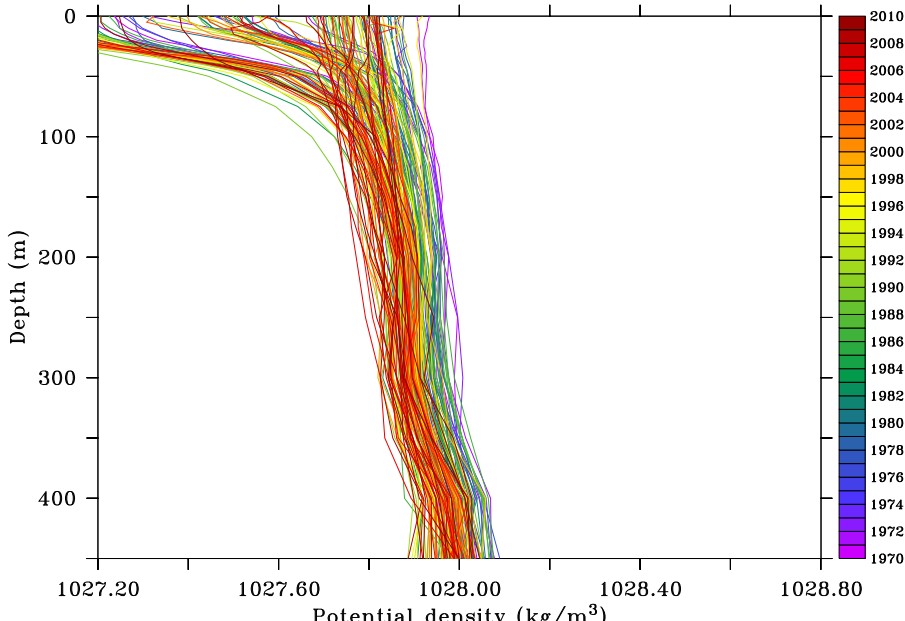

**Figure 9.** Seasonal averaged profiles of potential density on the most reliable area between 1970–2010.

associated with the "Great Salinity Anomaly of the 1980s" Dickson et al. (1988) is seen as a distinct maximum of salinities below 34.8. Finally, the potential density relative to the surface is shown in Fig. 12 where classes range between 1027.2 and 220  $1028.8 \, \mathrm{kg \, m^{-3}}$ with a step of $0.2 \, \mathrm{kg \, m^{-3}}$. The potential density does not display large changes on the long term, similarly to the conclusions made above by using profiles. However, water masses with densities above $1028.0 \, \mathrm{kg \, m^{-3}}$, associated with dense water production, has rarely exceeded 20 percent of the total water mass within the MRA after year 2000.

### 5.3 Ocean Heat Content

The Ocean Heat Content (OHC) change at the MRA is calculated following the method described in Boyer et al. (2007):

$$225 \quad OHC = \iiint \rho(t,s,p)c_p(t,s,p)\Delta t \, dx dy dz \tag{1}$$

where t and s are temperature and salinity averages at each location between 1970–2010, $\rho$ is the density of seawater averaged over 1970–2010 for each grid point, $c_p$ is the specific heat of seawater taken here as $3985 \, \mathrm{J \, kg^{-1} \, K^{-1}}$ (Hill, 1962) and $\Delta t$ is the temperature anomaly with respect to the averaged temperature on the reference period 1970–2010, that is 2.73° C.

Fig. 13a shows the OHC changes in the MRA between 1965–2015. The time series shows a positive trend of $5.043 \times 10^{16} \, \mathrm{J \, d^{-1}}$ 230  with a $R^2$ of 0.36, which is significant at a confidence level of 95%. We followed the Fisher–Snedecor test of significance described in Chouquet (2009) and Montgomery et al. (2012) augmented by a penalization of autocorrelation (Wilks, 1995). The temperature from the BSO extracted from ICES (https://ocean.ices.dk/iroc/#) is also shown. The correlation between the

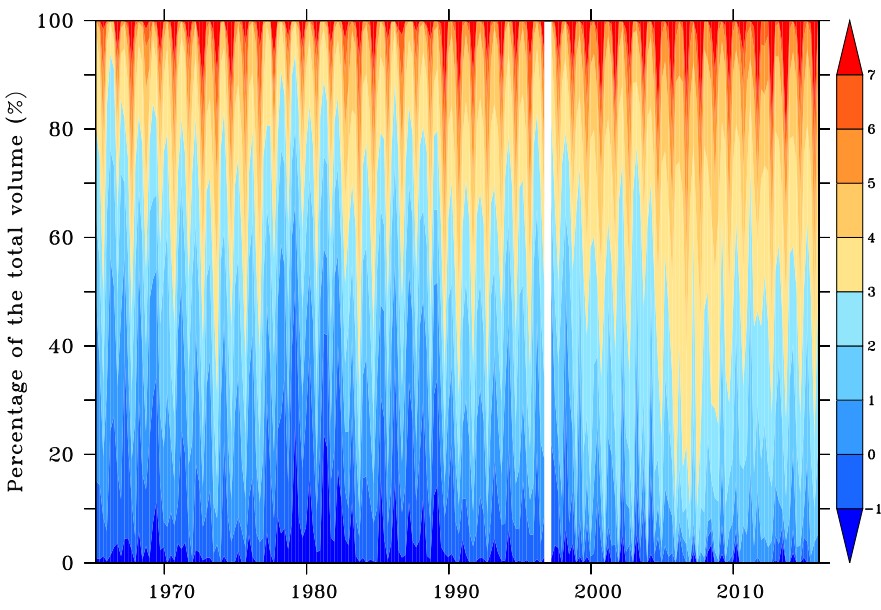

**Figure 10.** Volumetric temperature classes ranging from -1 to +7 °C in the most reliable area per season between 1965–2015.

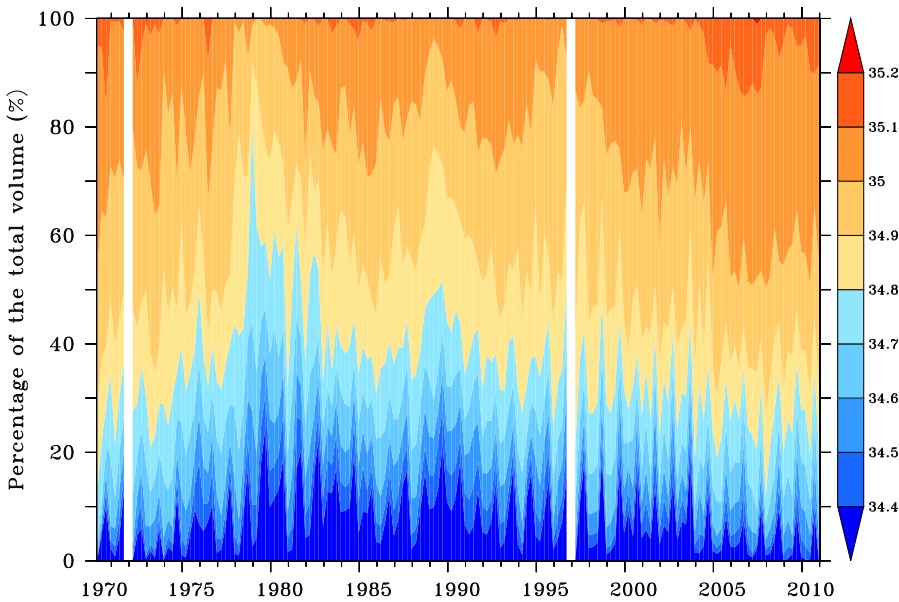

**Figure 11.** Volumetric salinity classes ranging from 34.4 to 35.2 in the most reliable area per season between 1970–2010.

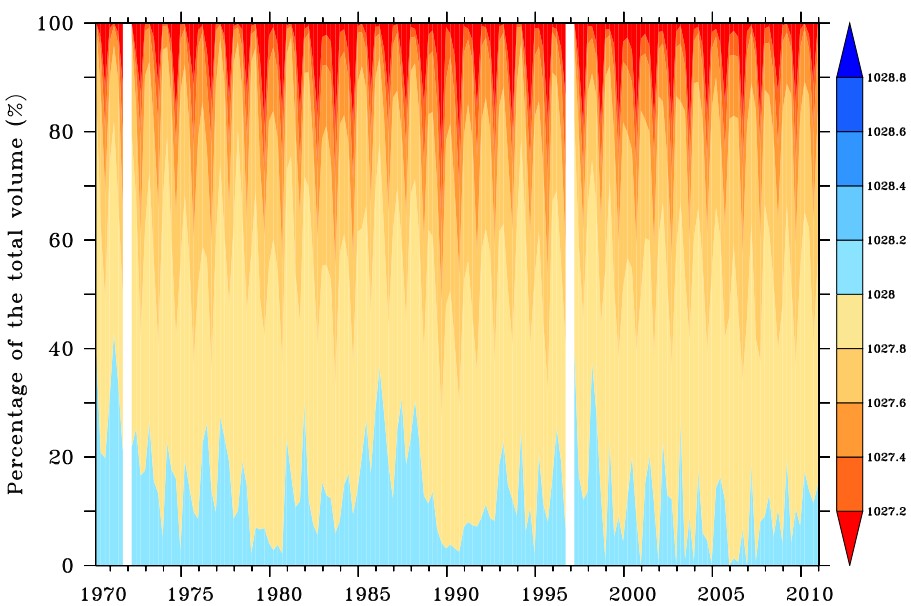

**Figure 12.** Volumetric potential density classes ranging from 1027.2 to 1028.8 kg m$^{-3}$ in the most reliable area per season between 1970–2010.

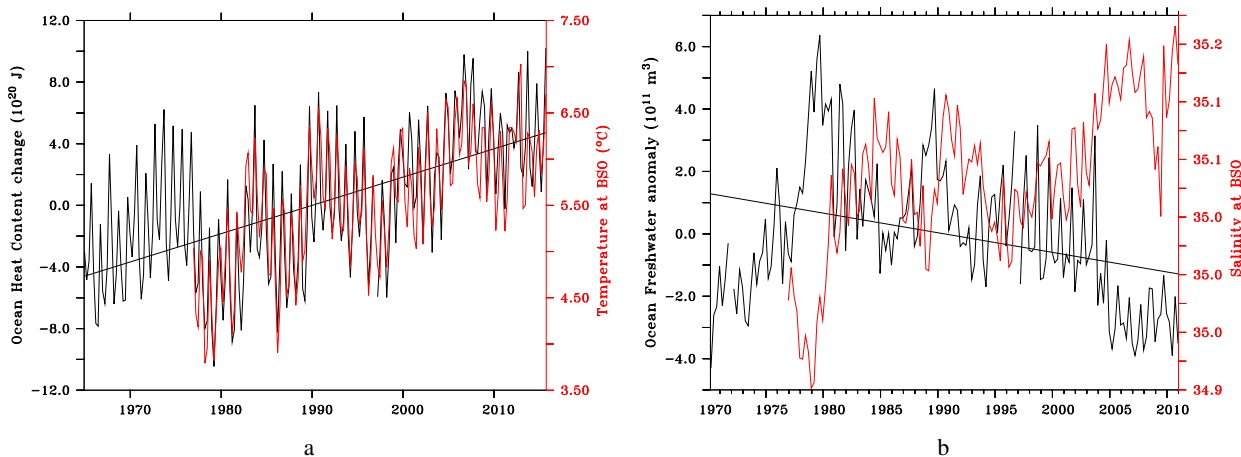

**Figure 13.** (a) Ocean heat content in the Most Reliable Area between 1965–2015, its linear trend (black) and temperature at the Barents Sea Opening. (b) Equivalent freshwater content in the Most Reliable Area between 1970–2010, its linear trend (black) and salinity at the Barents Sea Opening.

temperature at the BSO and the OHC is 0.89 (winter 1976– autumn 2015) and also significant at a confidence level of 95%, indicating that the temperature observed at the BSO is a reliable proxy for the OHC downstream in the southern part of the
Barents Sea.

## 5.4 Equivalent freshwater content

To investigate changes in salinity in the MRA, we use the Boyer et al. (2007) method to compute the Ocean FreshWater (OFW) anomaly.

$$OFW = -\iiint \frac{\rho(t,s,p)}{\rho(t,0,p)} \frac{\Delta s}{s + \Delta s} \, dx dy dz \tag{2}$$

where $\Delta s$ is the salinity anomaly with respect to averaged salinity on the reference period 1970–2010, that is 34.88, $\rho$ is the density of seawater at each grid point.

In Fig. 13b, changes in the OFW in the MRA are shown between 1970–2010. The slope is $-1.722 \times 10^7 \, \mathrm{m}^3 \, \mathrm{d}^{-1}$ with a $R^2$ of 0.11, which means the negative trend is not significant at a confidence level of 95%, although very close to the significance threshold. We followed the same method as for the OHC to examine the significance. The salinity at the BSO extracted from 245    ICES (https://ocean.ices.dk/iroc/#) is also shown. The correlation with the OFW between winter 1976–1977 and winter 2010–2011 is -0.57, also not significant but very close to the significance threshold.

## 6    Conclusions

This research provides a comprehensive atlas of temperature and salinity covering the whole Barents Sea on a regular grid, with an emphasis on its MRA. Although the *in situ* data is sometimes scarce in this part of the Arctic, we show here that physical 250    information can still be extracted from compiled databases by using a variational method minimising the expected errors on the resulting fields. These error fields can be used to exclude unreliable periods of areas, as shown by the examples of usage provided in this study. Besides, the regular grid facilitates the computation and the visualization of various metrics such as profiles, volumetric T-S diagrams or OHC and OFW.

The results of these examples are consistent with the recent "Atlantification" processes at the Barents Sea already observed 255    in the previous studies (e.g. Barton et al., 2018; Lind et al., 2018), that is warmer and more saline Barents Sea, even though our analysis only includes autumn when considering the whole Barents Sea. Concentrating on the MRA in the Barents Sea allowed us to analyze longer period (1965–2015) with all seasons included. The analyses showed similar results to the ones made for the whole Barents Sea, showing an overall positive temperature and salinity trend, that is +1.74° C between 1965–2015 and a salinity increase of 0.11 between 1990–2010. No clear trend was found in density due to the cancelling effects of both 260    temperature and salinity increase. This conclusion is supported by both vertical profiles and volumetric analysis. Finally, the computation of OHC and OFW are consistent with these conclusions as they show positive and negative trend, respectively, during the period 1965–2015 for the OHC and 1970–2010 for the OFW, although the latter trend is not significant. The measurements of temperature and salinity at the BSO are also consistent with the OHC and OFW variabilities. The code as well as the data are made available online (see Sections 4 and 7) to encourage further research on this topic.

# 7 Code and data availability

The Diva software we used for this research as well as its user guide are available here: https://github.com/gher-ulg/DIVA. The data is accessible at https://doi.org/10.21335/NMDC-2058021735 (Watelet et al., 2020).

*Author contributions.* Sylvain Watelet conducted the research and prepared the manuscript with contributions from all co-authors. Skagseth Øystein, Vidar Lien and Jean-Marie Beckers contributed in designing the research. Helge Sagen, Øivind Østensen, Vidar Lien helped preparing the data. Ivshin Viktor made possible the use of the Russian data. Jean-Marie Beckers helped with implementing DIVA.

*Competing interests.* The authors declare that they have no conflict of interest.

*Disclaimer.* The code is provided "as is", data quality as described by sources (e.g., WOD13). Diva is a software developed at the GeoHydrodynamic and Environmental Research (GHER, http://labos.ulg.ac.be/gher/) group at the University of Liège (https://www.uliege.be) and further developed for SeaDataNet scientific data products in JRA4 activities. Diva is copyright © 2006-2019 by the GHER group and is distributed under the terms of the GNU General Public License (GPL): http://www.gnu.org/copyleft/gpl.html

*Acknowledgements.* The authors would like to thank the NPI for preparing data. The DIVA development has received funding from: the European Union Sixth Framework Programme (FP6/2002-2006) under grant agreement n° 026212, SeaDataNet, Seventh Framework Programme (FP7/2007-2013) under grant agreement n° 283607, SeaDataNet II, SeaDataCloud and EMODNet (MARE/2008/03 - Lot 3 Chemistry - SI2.531432) from the Directorate-General for Maritime Affairs and Fisheries.

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
