# Peer review of "A volumetric census of the Barents Sea in a changing climate"

_Earth System Science Data, 2020_

## Referee Comment (RC1) · Anonymous Referee #1 · 24 Apr 2020

The manuscript is good prepared and covers the scope of the journal. I suggest to use a same time period (1970-2010) for the analysis. In the Fig.2, it is seen that the salinity data is available before 1970. The author should explain, why the data before 1970 was not used in the analysis. 1. Avoid such phrasal verb "First of all" or "indeed" in the scientific texts 2. Line 100: In the text is written that the temperature data was available for a time range 1965-2016. But in this line, the time range was changed to 1965-2015.

---

## Referee Comment (RC2) · Anonymous Referee #2 · 5 Jun 2020

Review of "A volumetric census of the Barents Sea in a changing climate" by Watelet et al.

The manuscript describes a data set of available temperature and salinity mapped to a regular grid in the Barents Sea. The mapping is not described in detail, but instead reference is made to a publicly available software DIVA. Some error estimates of the mapped fields are also provided with the data set, but the estimation procedure is not described. The mapped data is used to perform some basic analyses of temperature and salinity trends in the Barents Sea.

Technically, the manuscript is generally well written and easy to follow, some exceptions (and suggestions for improvement) are listed below. Some of the figures could be revised to improve the presentation and/or conciseness, see below, too.

[Figure]

From the "Aims and Scope" site of ESSD (https://www.earth-system-science-data.net/about/aims_and_scope.html) I cite: "Articles in the data section may pertain to the planning, instrumentation, and execution of experiments or collection of data. Any interpretation of data is outside the scope of regular articles. Articles on methods describe nontrivial statistical and other methods employed (e.g. to filter, normalize, or convert raw data to primary published data) as well as nontrivial instrumentation or operational methods. Any comparison to other methods is beyond the scope of regular articles."

In this sense the current paper is a data description paper, that should not contain "any interpretation" (but it does). As a methods paper it lacks the description of "nontrivial statistical and other methods". These issues aside, my main concerns are:

(1) There is a lack of detail in the description of the methods. For the generation of a data set from existing sets, I would have expected at least a rough explanation of the procedure beyond naming the software that has been used, for example, fundamental equations (objective function?) and constraints etc. In the same way, the error estimation method is named ("the clever poor man's method", something I have never heard of) with a proper reference (to paper in a journal that I don't have access to, embarrassingly enough), but that's all of the information that the reader gets. I think that for this type of journal and this type of derived data set at least a rough outline of the methods is appropriate.

(2) Some choices for gridding are not explained. For example, for a small region like the Barents Sea, why would one use a "lat-lon" grid instead of a proper projection with (nearly) constant grid spacing, or at least a scaled latitude coordinate (dlat = dlon*cos(lat)), so that the grid boxes are nearly square. With the chosen 0.1x0.1 deg grid, grid cells are elongated (making the mapping "anisotropic") and their volume varies by up to a factor of 2. With this choice, the "volumetric" analysis also contains this factor error of up to 2. As a consequence the volumetric t-s diagrams are not convincing.

[Figure]

(3) The manuscript claims to provide a comprehensive gapless data set, but then restricts the analysis to certain seasons, regions and years. The point of the entire data set as a whole is not clear to me, if even the authors of the data set don't want to use all of it. After all, the very applaudable inclusion of error estimates should allow to provide robust analyses (including error estimates), even when the underlying data is sparse and the corresponding errors are large. Some explanation seems in place, why we need this data set, if the even the authors don't trust all parts of it.

(4) I have issues with the use of "freshwater" and "equivalent freshwater content" in this manuscript. This strange and non-official convention (see the official definition of freshwater according to section 3.22 of the TEOS-10 Manual (IOC et al. 2010) as 1 minus the Absolute Salinity (in kg/kg)). It has never been clear to me, why one does not use salinity and salt content, which are straightforward and un-ambiguous quantities to describe the change in salinity in a given volume.

(5) I downloaded and superficially inspected the data. The salinity file contains many gaps in time, probably corresponding to the data availability in Fig3, but these gaps are not described in the text. From the text I would have expected annual mean of global mean fields with large error estimates instead (there are no error estimates for the gaps, either). This explain in part my issue (3) for salinity. I guess it makes little sense to fill the gaps where there are no salinity data available, but I think the text should clearly describe the gaps in the gridded data set.

(6) In the gridded salinity fields there are many unrealistically low numbers (~18 and even a large area of negative numbers down to -18 in timelevel 112 in the northeastern corner over the entire depth) that are not masked in the L1 and L2 versions of the fields (that have been masked according to the relative error thresholds). The temperature fields also contain many values near coastlines or in inlets that seem to be unrealistic, but naturally not as much as salinity, because (I guess) temperature values are generally closer to zero so that accidentally using a zero does not show up as badly as in a salinity field (which typical values around 33).

[Figure]

The last issue is severe and may make the data set not very useful to the community. All of the other concerns are not major by themselves, but together they will require a major revision of the manuscript and maybe of some of the computations. Hence my recommendation.

Minor comments and suggestion. I am attaching an annotated PDF with the same, but unrevised comments for better context.

page 1 l1: "Due to its location between the Norwegian Sea and the Arctic Ocean, the Barents Sea is one of the main pathways of the Atlantic Meridional Overturning Circulation."

Not sure if this statement is accurate: Why the location between NS and AO the cause of this? Rephrase.

l4: according to my dictionary, "prospect" is not a verb, except for "searching" as in "prospecting for gold". You probably mean forecast/predict or similar?

l16: "the most to the reduction" -> most of the reduction

page 2 l33 and elsewhere: I learned that abbreviations like "e.g." or "i.e." are to be used only within parentheses, but that they should be spelled out in regular text ("for example", and "that is").

page 3 l37: (e.g. hydrographic sections) also an example of limited coverage in time? unless you are talking about repeat sections

l43: "freshwater" see major comment (4) and also discussion of "freshwater" in recent paper by Schauer and Losch (2019), JPO, doi:10.1175/JPO-D-19-0102.1 or similarly Treguier et al (2014), OS, doi:10.5194/os-10-243-2014

l46: post -> after

ll52: DIVA is not introduced properly. Which is the proper reference? Rixen et al? or Troupin et al? In general the algorithm is hard to follow. I would not be able not

reproduce what you have done.

ll62: "then downgraded to a resolution of 1/8x1/8°", how? (and improve format for 1/8 x 1/8)

ll64: "The remaining data availability", is this per year? or per season?

l65: the plots 2 and 3 are difficult to read. Initially I even thought that some of the bars where stacked. Maybe fill the bars?

l66: "on" -> "for"?

l66: "four seasons", do you bin the data into the four seasons per year? Not clear from the text (and the figures 2 and 3)

page 5 ll72: improve description to make clear that there is a reference field for each season, i.e. 4 per year, etc. what is a "simple data average"? a horizontally averaged value that is use as a horizontally constant reference field/first guess?

ll84: "clever poor man's method, a good compromise between the computation time and the accuracy (see Beckers et al. (2014))" never heard of this, and unfortunately I don't have access to Beckers et al. (2014), please explain this method. Reference scheme: (see Beckers et al., 2014)?

ll85: "This error field on the analysis is then compared to the error on the first guess" -> This analysis error is then compared to the first guess error

ll86: "namely the relative error field which thus consists in a score comprised between 0 and 1" unclear, if this refers to the first guess error or the ratio of the first guess to the analysis error or some scaled difference between the two. Please be more specific.

l88 would be the true field how can be know the true field?

l92: "The statistical parameters and the analysed fields masked when the relative error exceeds 0.3 or 0.5" awkward, please rephrase.

l95: "gave" -> provided

l96: "from mid–2000s than previously" rephrase and fix grammar

page 6 l99: "uncertainties on the Atlas" uncertainties of the atlas data (not clear why you spell atlas suddenly with a capital A)

l99: "The BS has a varying data coverage" -> The data coverage in the BS varies from year to year.

l100: "relative" wouldn't the absolute errors be more instructive? Now these are errors relative to very small temperature values (close to zero)

The entire error estimation is unclear to me.

l100: cut "BS"?

l102: "averaged on all layers" -> averaged over all layers

l103: minimum -> minimal

ll104: "For this reason, we decided to focus on the autumn only when considering the whole BS." Make clear to which extent this is a limitation of your analysis.

l107: PSU there is not "PSU" and salinity has no units, e.g. absolute salinity has g/kg, but even "regular" salinity is unitless

l110: "here not taken into account" why not? Apparently a factor up to two is involved. Is that a problem? Having a converging lat-lon grid for such a small area is questionable to begin with. Why this choice?

ll113: "due to the cancelling effects from the increasing haline contraction and thermal expansion on density" -> due to the cancelling effects of increasing haline contraction and thermal expansion on density

l116: "at most" ???, the most?

[Figure]

page 7 Caption Fig6: "Average relative error on the Barents Sea for temperature"

-> "Average relative temperature error in the Barents Sea" (and similar for salinity)

fig6 caption: "seasons" it is a function of time, not of seasons (the labels are years)

page 8 Fig7a caption "Average volumetric T-S diagrams during 1994–1998 and 2006–2010" is unclear, rephrase (the version in the text is clear)

l118: "it is clear the error" insert "that"

l119: "This strengthens the reliability of the observed T-S changes." This is not clear to me, large uncertainties mean few data points, changes cannot be detected with few data points, so many changes may have gone unnoticed?

page 9 l128: we focus on the periods

l131: "One way of studying changes in temperature and salinity in the MRA is to look at the vertical dimension."

I would remove this sentence. No additional information and the phrasing is not very "scientific" (e.g. you can "look at a piece of art", or "look at me, when I am talking to you", but I would study/inspect/analyse/take into account the vertical dimension).

l131: "Temporal . . ." The temporal evolution . . . is shown . . .

Fig. 9, 10, 11: consider a different presentation of the data, e.g. a Hovmøller-like plot as in Fig 12 and 13 (except depth on the y-axis), the current plots are difficult to read. Maybe you can find a good way of combining Fig 9 to 13 in two or three panels. Now they take up a lot of space for limited information.

page 9 l134: raise -> increase

l134: here and elsewhere: I am not a friend of abbreviations and I would consider spelling out Barents Sea every time you use "BS".

page 10 Section 6.3 It is not clear that the volumetric changes in T/S and density

provide new information over the profiles (it get's warmer, salinity is ambiguous and density doesn't change very much), so the use of this section is not clear (and this has implications for the title of the paper, so I would ask for a better explanation of the volumetric t-s diagrams, etc.)

eq(1) can only be a Ocean Heat Content (OHC) change, because deltaT is the change of temperature relative to a reference.

page 13 l150, 153: SI units are not supposed to be in italics

l153: "significant to the 95% level" not sure if this is the appropriate formulation

eq(2) [and to some extend eq(1)] what is delta s : s_ref-s? if so, then delta s/ (s+delta s) = s_ref-s/s_ref?

It is not clear what this EFWC is supposed to be. The proper (e.g. TEOS10) definition of freshwater is ocean-water minus salt (i.e. 1-s). In this sense, eq(2) can only be some fractional freshwater content (and just because it has been called freshwater before doesn't make it right). Because eq(2) depends on a reference salinity (the value of which is not even provided here), it is impossible to related the calculated numbers to anything else. Also the choice of reference (be it the mean as in your case or some arbitrary value) makes a difference in the time series. See Schauer and Losch (2019), their Figs3+4 for a simple illustration, also the discussion in Treguier et al (2014)

Similarly the OHC in eq(1) depends on the reference (and the units, do you use degC or Kelvin?). In the OHC case one can argue that everyone in oceanography uses degC and a reference of 0degC to compute OHC so that the ambiguity problem goes away (see McDougall, 2003, doi:10.1175/1520-0485(2003)033<0945:PEACOV>2.0.CO;2). Here the reference appears to be the mean temperature resurrecting the same problem as for the salinity anomaly/fractional freshwater.

161: SI units not in italics

l163: "For both OHC and EFWC trends significance, we followed the Fisher–Snedecor

test described in Chouquet (2009) and Montgomery et al. (2012) augmented by a penalization of autocorrelation (Wilks, 1995)"

this information should have come earlier, also: "For both OHC and EFWC the significance of the trends was determined following ..."

page 14 l165: any idea or comment why the salinity trend at BSO is opposite to "EFWC"? Maybe because of the minus sign in the definition? Wouldn't it make more sense to reverse the sign in the plot to illustrate the correlation?

This also goes back to my point of eq(2): Using salt content (integral over salinity) would be a less ambiguous measure and would yield itself much more easily to physical interpretation.

page 14 Conclusions

the conclusions are weak, but since this is a data product, there may not have to be strong conclusions about the physical interpretation. I would focus on the presentation of the data in the conclusion.

l169: on this part -> in this part

l169: "much" replace by "some" or remove

l170: "provided a variational method minimising the expected errors on the resulting fields is used" I don't think that this research shows that this method is required for the analysis. To be able to draw this conclusion I would like to see why it is impossible to extract physical information from sparse data without this interpolation method. Please rephrase.

Please also note the supplement to this comment:
https://www.earth-syst-sci-data-discuss.net/essd-2020-70/essd-2020-70-RC2-supplement.pdf

[Figure]

**Supplement:**

[revised manuscript text omitted]

---

## Referee Comment (RC3) · Anonymous Referee #3 · 6 Jun 2020

Review of the manuscript "A volumetric census of the Barents Sea in a changing climate", by Watelet et al.

This paper presents a novel temperature and salinity datasets that is presented on a regular grid for the Barents Sea, which are "constructed" from the available datasets. It also presents basic analysis of the thermohaline trends that are showing the state of the Barents Sea. The manuscript is well written and is easy to follow. For these reasons, I think that presented manuscript has a potential for the publication in the ESSD. However, I have some concerns regarding the manuscript, especially when it comes to the description of the used methods to "construct" the maps of temperature and salinity. In the manuscript, the Authors mention the DIVA software package, however I am missing the details on the used methods and the existing data. My other main concern

is the coverage (in space and time) of the used thermohaline data, as well as for the final gridded maps, that were not clearly described in the manuscript, and for which I think that is important for the reader to understand your choice of analysis that are limited to seasons, years and/or regions. Altogether, I recommend this manuscript for the publication after the major revisions.

The main concerns from my side are:

1. In the manuscript, I find that Section 2 is poorly described, which could lead the readers to miss the important details of the used data. You could provide a Figure of the Barents Sea with the locations of the used in situ data, and others. Are these data measured regularly (for instance on a monthly basis)? In addition, the Authors could also provide the references for the WOD13 data, PINRO CTD data and the NPI data.

2. The description of the method could include more details. Aside from the DIVA package, you mentioned several techniques that you used to create the gridded data with the estimated error fields, however, I am missing the details of the methods. For instance, until now I have never heard of the "clever poor man's method", and I couldn't get access to a reference paper. You could provide additional information about the used methods, since this is the main work of your manuscript.

3. Also, just by reading the Sections 3, I find it a bit confusing, which is a bit disappointing, considering that this Section is containing the main core of your paper. First of all, you should make it clear for the readers what are the input data and what are the resulting gridded maps. Similar to my previous comment, I am missing the description of the datasets. What is the coverage of the original datasets, and what is for the resulting gridded maps (in space and time)? Are the resulting maps on a monthly basis? In L57: you mention that "the ODV spreadsheets were vertically interpolated onto 23 depths...", however, I am missing the information on how the spreadsheets were constructed. Do they contain only the original data, and if so, what does one spreadsheet represent and how many data does it contain? Also, you mentioned that you performed

the analysis on four seasons (L67), and that you constructed the data for these analysis by using an 11-year windows (for which I am not sure whether you averaged only the corresponding seasons during these 11 years, or the whole year). At this point, I was not sure whether this analysis is performed on the data that are showed in the T-S diagrams? I have to admit, I was bit confused reading this Section, so my suggestion is to rewrite it in a form that is more precise, and providing more details on the used methods, including the methods used for estimation of the errors.

4. Regarding the basic analysis of the gridded thermohaline fields, I find these Sections too descriptive. You should provide numbers when making statements, i.e. whenever you use phrases such as "increase", "decrease", "trend", etc., you should a give a value by "how much". Some examples are given bellow in the detailed comments

Minor detailed comments and suggestions:

Figures:

1. Most of the Figures are lacking the descriptions. Each of the Figure should contain detailed description of what it is showing. For instance, in Fig.2 and 3 it is not clear for me what the time-step for the data is. On Fig. 4 and 5, you should provide explanation for the values on the colorbar. On Fig. 7 should be mentioned which type of data is shown.

2. In Fig6a the y-axis: "Averaged relative temperature error" or "Averaged relative error for temperature". Same for the salinity (Fig6b)

L15 and elsewhere: I don't understand the usage of BS acronym. However, if you chose it, you should be more punctilious while using it. In this line you mention "Barents Sea", without the acronym, and you introduce it in the L16. Throughout the manuscript you are sometimes using "Barents Sea", and sometimes "BS". This should be corrected also at the Figure 1. Same comment applies for all the other used acronyms.

L17: use the apostrophes when mentioning Atlantification. As far as I know, this is not
a name for a physical process, even though the readers do understand the meaning of the phrase.

L18: what are the "both physical conditions"? Perhaps you can exclude "such as", since it implies that there are more than two

L18 and 19: as well as on biological and marine ecosystem

L30: varies between seasons and years, especially during winter and spring

L31: . . . or concentrated at fixed sections.

L32: . . . sea surface temperature, and recently sea surface salinity

L33: Don't use "E.g." at the beginning of the sentence. Instead you could say "For example,. . ."

L34: . . . the Arctic that shows temperature increase for the period... What was the increase?

L35: . . . between the two periods: 1979-1995 and 1996-2012.

L36: . . .property changes,

L36: I find the phrase "in situ" differently written at several places. Sometimes it is "in situ", and sometimes is "in-situ". Try to be consistent when using phrases. I found in many previous papers written in curves "in situ"

L36 and 37: . . . often have disadvantages of. . .. and/or time (sometimes it could be both)

L37 and 38: Please rephrase the sentence, it doesn't sound grammatically correct.

L39 and 40: Does the seasonal temporal resolution mean a 4-month averages? You should be more concise here, as well as in Section 3 (See my comment 3 in the main concerns)

L40: "based on all available observations". Does this also include satellite data? It is not that clear in the Section 2.

L53: . . . in situ data using a Variational. . .

L63 to 65: To which data are you referring here, original or the ones already interpolated on the 23 layers? It is not easy to follow this Section. At this point I am understanding that the Fig 2 and 3 are showing the interpolated data. However, L104 suggests that these Figures show original measurements. I find it a bit confusing.

L67: You should define the periods for the 4-month averages, just to be more precise. Later on, you use "autumn", and you never defined that season. Even though it is self-clear, I find it better to be as much as precise as possible when writing a scientific paper. My suggestion "November to January (autumn), etc."

L96: from the mid-2000s. Also, to what period do you refer when saying "than previously"?

L97: Why is the reason of choosing these exact 5-years periods? Also, could you give an exact number of the data used in the analysis?

L100: Why didn't you include year 2016 for the estimation of the error fields?

L102 and 103: The relative error field averaged through all the layers for each variable and season is shown in Fig. 6

L105: there is no need to say "only when considering the whole BS", since this is the only analysis that is considering the whole BS.

L106: Volumetric T-S changes for both periods were carried out by summing all the pixels falling inside the T-S classes . . . having a step of . . .

L106: Does the data only correspond to the autumn data? You should also mention this in the description of the Fig. 7

L112 to 113: You should mention that this increase implies only to autumn. Also, this is a good example of descriptive sentence. You should provide averages: "the increase in temperature and salinity is clear, by XX C and XX PSU in average"

L115: In Section 3 u stated that the reference fields were defined by an 11-year window, and here you say that you used a 10-year window. Please provide additional explanation.

L117 to 119: ... weighted by the layer... for periods 1994-1998 and 2006-2010. It is clear that the error is much lower on the T-S classes showing larger changes, which are ... This strengthens the reliability of the observed autumn T-S changes in the BS.

L122: Rephrase the sentence and correct the grammar

L125 to 127: If the errors are not shown, you need to state so

L127. This advantage allows us to analyze all the seasons at the MRA, in contrast to the whole BS. Here we focus on the periods: 1965-2015 for the temperature and 1970-2010 for the salinity. Also, add a sentence as an explanation of using these periods.

L128: You state that the years 1996-1997 is having a lack of data for the temperature at the MRA, which is showing the lowest errors. However, those two years are within the 5-years period that you used in the analysis of the previous Section, where you showed an increase in temperature. At this point, I am not convinced in the reliability of the previous analysis, and even more I still don't understand the choice of those 5-year periods for that analysis. Could you please explain?

Subsection 6.2: Why did you choose to show vertical seasonal thermohaline profiles, all in one plot? The figures are a bit "messy", showing all seasons, and it is not that clear to depict the trends. Instead of the profiles, you could show surface plots with the estimated trends for three different averaged layers (0-50, 50-300, and 300 to bottom). Other choice could be (a), (b), (c), and (d) for the profiles, where you could separate 4 seasons.

L131: I don't understand the point of this sentence. You should remove it.

L132 to 134: I also find this descriptive. What is the averaged values for the temperature increase? Also for the salinity, "unambiguous raise between the 90s and the 2000s", how much? ... "similarly to the observation made for the whole BS", I am missing a reference here.

Subsection 6.3.: Provide additional information on how you estimated the volume of water? What exactly are the Figures 12, 13 and 14 showing? Once again, I find this paragraph too descriptive. When using phrases such as "increase", "decrease", "trend", etc. you should a give a value by "how much".

L139: I don't understand the phrase "classes" in this sentence. Define the "classes".

L139: The calculations. . .

L145: . . . similarly to the conclusions made in Section 6.2.

Fig 12, 13 and 14: Are the diagrams showing the sums of the volumes in a whole seasons? You should state that in the description of the Figures

L146: . . . at the MRA. . .

L151: Define "reference period"

L152: In the Formula (1), OHC is dependent on the density changes, which is dependent on both temperature and salinity changes. How could you estimate OHC value for the period outside 1970-2010? Even in the L149 you stated that t and s are between 1970-2010. Could you please explain?

L154: Is the correlation significant? From the Figure 15a, it clearly is, but it is better to add it in the sentence as well.

L155: . . . at the BSO

L161: In Fig. 15b changes in the EFWC . . .

L163 and 164: To which threshold do you refer? The sentence on the choice of the significance tests should be stated before.

L165: Is the correlation significant? If yes, could you give a sentence in explanation on why the correlation is negative, similar to the one you gave for the OHC and temperature positive correlation? Also, it would be interesting to know what caused an extreme salinity decrease during the late 1970s and early 1980s, evident in both EFWC increase (Fig15b) and in the percentage of the total volume (Fig13). Are there any references for that?

L170: Rephrase the sentence

L171 and 172: "The results are consistent with the recent "Atlantification" processes at the BS already observed in the previous studies, i.e. warmer and more saline BS, even though our analysis only includes autumn when considering the whole BS". Also, I am missing a references for the previous studies

L172 to 175: Concentrating on the MRA in the BS allowed us to analyze longer period (1965-2015) with all seasons included. The analyses showed similar results to the ones made for the whole BS, showing an overall positive temperature and salinity trend (with numbers!), while . . . cancelling effects of both temperature and salinity increase.

L176 and 177: . . .these conclusions as they show positive and negative trend, respectively, during the period 1965-2016. I am a bit concerned here. As I stated before in my comments, I find this period suspicious. Moreover, EFWC was estimated only for the period 1970-2010. In addition, I don't think that EFWC trend is significant, since R2 is only around 12%.

Please also note the supplement to this comment:
https://www.earth-syst-sci-data-discuss.net/essd-2020-70/essd-2020-70-RC3-supplement.pdf

---

## Author Response (AR1)

**Last update (23/07/2020):** The replies below were only partly included in the manuscript due to other changes linked with our replies to the other referees. Our answers remain valid, but they are only included in the paper if not purposeless.

*Comment:*

Anonymous Referee #1

The manuscript is good prepared and covers the scope of the journal. I suggest to use a same time period (1970-2010) for the analysis. In the Fig.2, it is seen that the salinity data is available before 1970. The author should explain, why the data before 1970 was not used in the analysis. 1. Avoid such phrasal verb "First of all" or "indeed" in the scientific texts 2. Line 100: In the text is written that the temperature data was available for a time range 1965-2016. But in this line, the time range was changed to 1965-2015.

*Reply:*

**We would like to thank the Referee #1 for the time spent in the review of our work and its positive assessment. Hereafter, we answer in bold to the comments:**

Referee 1: I suggest to use a same time period (1970-2010) for the analysis. In the Fig.2, it is seen that the salinity data is available before 1970. The author should explain, why the data before 1970 was not used in the analysis.
**Reply: There is no salinity data available on the Barents Sea before 1970 except a few in Autumn, as shown on Fig. 3., while the Fig. 2 only refers to temperature. This is confirmed on the error graphs of both the Barents Sea (Fig. 6b) and the Most Reliable Area (Fig. S2). As the temperature data is also present before 1970 and after 2010, we discarded only 2016 for our analyses on the MRA, due to the huge error estimate this particular year. The temperature analysis between 1970-2010 is however easy to visualise on our graphs, so we do not think removing 1965-1969 and 2011-2015 is useful.**
**For greater clarity, we will include this reasoning in the manuscript.**

Referee 1: Avoid such phrasal verb "First of all" or "indeed" in the scientific texts
**Reply: We will remove these phrasal verbs from the manuscript.**

Referee 1: Line 100: In the text is written that the temperature data was available for a time range 1965-2016. But in this line, the time range was changed to 1965-2015.
**The atlas is available on 4 seasons between 1965 and 2016, which are defined as follows: November to January for winter, February to April for spring, May to July for summer, and August to October for autumn. The first season is thus November 1964 to January 1965, the last being August to October 2016. Considering this, we preferred to make the average between spring 1965 and winter 2015-2016, in order to have only full years running from February to January. Otherwise, there would be two**

biases involved: no data in November and December 1964 and no data for November and December 2016.

For greater clarity, we will include this reasoning in the manuscript.

**Reply to Interactive comment on "A volumetric census of the Barents Sea in a changing climate" by Sylvain Watelet et al.**

We would like to thank the Referee #2 for the time spent in the review of our work. Hereafter, we answer **in bold** to the comments.

Comment and reply:

Anonymous Referee #2

The manuscript describes a data set of available temperature and salinity mapped to a regular grid in the Barents Sea. The mapping is not described in detail, but instead reference is made to a publicly available software DIVA. Some error estimates of the mapped fields are also provided with the data set, but the estimation procedure is not described. The mapped data is used to perform some basic analyses of temperature and salinity trends in the Barents Sea.

Technically, the manuscript is generally well written and easy to follow, some exceptions (and suggestions for improvement) are listed below.

**We would like to thank the Referee #2 for this positive assessment of our work.**

Some of the figures could be
revised to improve the presentation and/or conciseness, see below, too.

From the "Aims and Scope" site of ESSD (https://www.earth-system-science-data.net/about/aims_and_scope.html) I cite: "Articles in the data section may pertain to the planning, instrumentation, and execution of experiments or collection of data. Any interpretation of data is outside the scope of regular articles. Articles on methods describe nontrivial statistical and other methods employed (e.g. to filter, normalize, or convert raw data to primary published data) as well as nontrivial instrumentation or operational methods. Any comparison to other methods is beyond the scope of regular articles."

In this sense the current paper is a data description paper, that should not contain "any interpretation" (but it does). As a methods paper it lacks the description of "nontrivial statistical and other methods".

**We acknowledge this point, and we have written an introduction to Section 5 stating that the analysis presented provide examples of usage of the data product and its features, rather than an in-depth analysis of the climatic conditions. The text reads:**
***"In the following sections we demonstrate how the error field provided in the atlas can be utilized to objectively limit the data in time or space before applying the desired analysis. Moreover, we give some examples of possible usages of the atlas product.".***

These issues aside, my main concerns are:

(1) There is a lack of detail in the description of the methods. For the generation of a data set from existing sets, I would have expected at least a rough explanation of the procedure beyond naming the software that has been used, for example, fundamental equations (objective function?) and constraints etc. In the same way, the error estimation method is named ("the clever poor man's method", something I have never heard of) with a proper reference (to paper in a journal that I don't have access to, embarrassingly enough), but that's all of the information that the reader gets. I think that for this type of journal and this type of derived data set at least a rough outline of the methods is appropriate.

**We added this text on the DIVA method: "In practice, the aim of the VIM is to minimize the following cost function J:**
**[equation, see manuscript]**
**where the N d observations d j are used to reconstruct the analysed field φ and with**
**[equation, see manuscript]**
**where α0 penalizes the field itself (anomalies with respect to a reference field, e.g., a climatological average), α1 penalizes gradients (no trends), α2 penalizes variability (regularization), and μ j penalizes data-analysis misfits (objective analysis) (Troupin et al., 2016). Unless specified otherwise, we always use the command line version of DIVA in this study. This version comes with the full set of options, for instance regarding the optimization of the statistical parameters later used in the analyses.". Regarding the clever poor man's error, we added this explanation: "The poor man's error is computed by analysing a "data" vector with unit values and is very cost-effective (Troupin et al., 2010), but the error field is too optimistic. It is shown that using the same method with a correlation length divided by a factor ~1.7 requires a similar computation time and yields a more realistic estimate of the error, that is, the clever poor man's error."**

(2) Some choices for gridding are not explained. For example, for a small region like the Barents Sea, why would one use a "lat-lon" grid instead of a proper projection with (nearly) constant grid spacing, or at least a scaled latitude coordinate (dlat = dlon*cos(lat)), so that the grid boxes are nearly square. With the chosen 0.1x0.1 deg grid, grid cells are elongated (making the mapping "anisotropic") and their volume varies by up to a factor of 2. With this choice, the "volumetric" analysis also contains this factor error of up to 2. As a consequence the volumetric t-s diagrams are not convincing.

**Other atlas products, such as the WOA, are provided on regular lat-lon grids, as well as most operational ocean models. Hence, it makes some of the usages more straightforward. We included this reasoning in the manuscript. We agree that the previous version of the paper included this factor error up to 2, and we now use a weighting function to fully overcome this issue.**

(3) The manuscript claims to provide a comprehensive gapless data set, but then restricts the analysis to certain seasons, regions and years. The point of the entire data set as a whole is not clear to me, if even the authors of the data set don't want to use all of it. After all, the very applaudable inclusion of error estimates should allow to provide robust analyses (including error estimates), even when the underlying data is sparse and the corresponding errors are large. Some explanation seems in place, why we need this data set, if the even the authors don't trust all parts of it.

**While the dataset is comprehensive, it is not without gaps or regions in space or time with less extensive data coverage (which is usually the case for spatially distributed in-situ datasets). This has now been made clear with the inclusion of additional information about data sources (as requested by the reviewers), including information that the data generally covers the ice-free parts of the Barents Sea, which limits the data coverage in winter. In addition, in Section 5 we utilize the error field to identify areas that are least affected by sparse data sampling or gaps as examples to guide the user in choosing regions with tolerable error estimates. Furthermore, we have included a short paragraph in Section 4 on the usefulness of gridded datasets in general and the presented dataset in particular.**

(4) I have issues with the use of "freshwater" and "equivalent freshwater content" in this manuscript. This strange and non-official convention (see the official definition of freshwater according to section 3.22 of the TEOS-10 Manual (IOC et al. 2010) as 1 minus the Absolute Salinity (in kg/kg)). It has never been clear to me, why one does not use salinity and salt content, which are straightforward and un-ambiguous quantities to describe the change in salinity in a given volume.

**As mentioned in the answer to the reviewer's general point, our objective is not an in-depth analysis of the Barents Sea climatic conditions, but rather to show examples of the atlas' possible usages, and in this specific case to visualize the temporal variability in the freshwater or salinity content of the Barents Sea. For a thorough investigation of the Barents Sea salt budget we agree that the suggested approach would be more appropriate.**

(5) I downloaded and superficially inspected the data. The salinity file contains many gaps in time, probably corresponding to the data availability in Fig3, but these gaps are not described in the text. From the text I would have expected annual mean of global mean fields with large error estimates instead (there are no error estimates for the gaps, either). This explain in part my issue (3) for salinity. I guess it makes little sense to fill the gaps where there are no salinity data available, but I think the text should clearly describe the gaps in the gridded data set.

**We have added this information in Section 4 as: "As shown in Fig. 2 and 3, there are several seasons with data gaps. In such cases, the atlas only contains a missing value, for both the analysis and the error field. The data gaps for salinity are mainly found before 1970 and after 2010, while the temperature has only exceptional data gaps. Between 1970--2010, there are data gaps in the salinity atlas during the 1971—1972 winter period and in both temperature and salinity atlas during the 1996--1997 winter period. Besides, other gaps appear sometimes in the deepest layers. In Section \ref{error}, we explain how to make use of the error field to take into account the data**

**coverage before applying any analysis."**

(6) In the gridded salinity fields there are many unrealistically low numbers (~18 and
even a large area of negative numbers down to -18 in timelevel 112 in the northeastern
corner over the entire depth) that are not masked in the L1 and L2 versions of the fields
(that have been masked according to the relative error thresholds). The temperature
fields also contain many values near coastlines or in inlets that seem to be unrealistic,
but naturally not as much as salinity, because (I guess) temperature values are gener-
ally closer to zero so that accidentally using a zero does not show up as badly as in a
salinity field (which typical values around 33).

**We agree with the Referee #2 that the atlas included several unrealistic values, especially for salinity. After a thorough analysis of the issues, we decided to fully recompute the atlas as well as the derived results and figures. We have decided to be more severe on the data quality by adding a range check for both temperature and salinity data and only using quality flags corresponding to good data. Besides, we capped the signal-to-noise ratio to 3 for temperature to avoid overfitting, and we applied a logit transformation on temperature to ensure the analysis does not generate temperature below -1.9° C. Finally, several fjords were removed before the calculations to avoid unrealistic extrapolation. All these changes are described in the manuscript.**

The last issue is severe and may make the data set not very useful to the community.
All of the other concerns are not major by themselves, but together they will require a
major revision of the manuscript and maybe of some of the computations. Hence my
recommendation.

Minor comments and suggestion. I am attaching an annotated PDF with the same, but
unrevised comments for better context.

page 1 l1: "Due to its location between the Norwegian Sea and the Arctic Ocean,
the Barents Sea is one of the main pathways of the Atlantic Meridional Overturning
Circulation."
Not sure if this statement is accurate: Why the location between NS and AO the cause
of this? Rephrase.

**We rephrased to: "The Barents Sea, located between the Norwegian Sea and the Arctic Ocean, is one of the main pathways of the Atlantic Meridional Overturning Circulation."**

l4: according to my dictionary, "prospect" is not a verb, except for "searching" as in
"prospecting for gold". You probably mean forecast/predict or similar?

**We replaced it by "investigate".**

l16: "the most to the reduction" -> most of the reduction

**Done.**

page 2 l33 and elsewhere: I learned that abbreviations like "e.g." or "i.e." are to be used only within parentheses, but that they should be spelled out in regular text ("for example", and "that is").

**Done.**

page 3 l37: (e.g. hydrographic sections) also an example of limited coverage in time? unless you are talking about repeat sections

**We replaced the above expression by "repeated hydrographic sections".**

l43: "freshwater" see major comment (4) and also discussion of "freshwater" in recent paper by Schauer and Losch (2019), JPO, doi:10.1175/JPO-D-19-0102.1 or similarly Treguier et al (2014), OS, doi:10.5194/os-10-243-2014

**See our answers to the major and other minor comments on this topic.**

l46: post -> after

**Done.**

ll52: DIVA is not introduced properly. Which is the proper reference? Rixen et al? or Troupin et al? In general the algorithm is hard to follow. I would not be able not reproduce what you have done.

**We have now included more details on DIVA in this Section. All the cited references are useful to describe DIVA as they are covering different aspects of the method. See also the Section "code and data availability" for a link towards the software and its user guide.**

ll62: "then downgraded to a resolution of 1/8x1/8∘", how? (and improve format for 1/8 x 1/8)

**We adapted the sentence as follows: "This bathymetry was then smoothed to a resolution of 1/8° by using a 2D convolution low-pass filter followed by a linear interpolation to avoid too complex shapes when computing the coastlines for each depth level."**

ll64: "The remaining data availability", is this per year? or per season?

**Per season, we updated the text and captions for clarity.**

l65: the plots 2 and 3 are difficult to read. Initially I even thought that some of the bars where stacked. Maybe fill the bars?

**We agree and changed the plots accordingly, thank you for the suggestion.**

l66: "on" -> "for"?

**Done.**

l66: "four seasons", do you bin the data into the four seasons per year? Not clear from the text (and the figures 2 and 3)

**The data are binned into each season before being analysed. We changed the text into "the objective is to perform one analysis for each season" for clarity.**

page 5 ll72: improve description to make clear that there is a reference field for each season, i.e. 4 per year, etc. what is a "simple data average"? a horizontally averaged value that is use as a horizontally constant reference field/first guess?

**We added "The horizontal average is used as a constant first guess when creating the reference fields. Therefore, 4 reference fields are generated per year, that is one per season.".**

ll84: "clever poor man's method, a good compromise between the computation time and the accuracy (see Beckers et al. (2014))" never heard of this, and unfortunately I don't have access to Beckers et al. (2014), please explain this method. Reference scheme: (see Beckers et al., 2014)?

ll85: "This error field on the analysis is then compared to the error on the first guess" -> This analysis error is then compared to the first guess error

**Done.**

ll86: "namely the relative error field which thus consists in a score comprised between 0 and 1" unclear, if this refers to the first guess error or the ratio of the first guess to the analysis error or some scaled difference between the two. Please be more specific.

**We adapted the sentence as follows: "This analysis error is then compared to the first guess error, and the ratio of those errors yields the relative error field which thus consists in a value between 0 and 1."**

l88 would be the true field how can be know the true field?

**Obtaining a zero error field would be possible if you had full coverage of observations and each observation with zero observational (including representativity) errors. In that case a zero error field would mean you perfectly trust your observations to be representative of the climatology you are analyzing. Of course that will never happen and this is why you don't get zero errors. A relative error of 1 on the other hand means that your data did not provide any information to the analysis (either because the data error is considered very high and/or because you are in regions where there are no data).**

l92: "The statistical parameters and the analysed fields masked when the relative error exceeds 0.3 or 0.5" awkward, please rephrase.

**We extended and splitted the sentence into: "The statistical parameters (correlation length and signal to noise ratio) and the analysed fields restricted to the most reliable areas are also available. These latter analyses are masked if the relative error exceeds 0.3 or 0.5."**

l95: "gave" -> provided

**Done.**

l96: "from mid–2000s than previously" rephrase and fix grammar

**We amended the sentence as: "[...] suggesting a warmer and saltier northern BS since the mid--2000s"**

page 6 l99: "uncertainties on the Atlas" uncertainties of the atlas data (not clear why you spell atlas suddenly with a capital A)

**Done, we removed the capital A everywhere in the manuscript as well.**

l99: "The BS has a varying data coverage" -> The data coverage in the BS varies from year to year.

**Done.**

l100: "relative" wouldn't the absolute errors be more instructive? Now these are errors relative to very small temperature values (close to zero)
The entire error estimation is unclear to me.

**The relative error field is a ratio of two absolute error fields, its values are thus not necessarily higher if the temperature is closer to zero. We think that this field is more instructive than the absolute error field, since it allows the visualisation of the reliable vs unreliable zones through a value [0-1] that shows the added value of the in situ data. However, the absolute error fields are also provided in the atlas.**

l100: cut "BS"?

**We decided to keep "whole BS" since there is value in mentioning we do not restrict this analysis to the northern BS.**

l102: "averaged on all layers" -> averaged over all layers

**Done.**

l103: minimum -> minimal

**Done.**

ll104: "For this reason, we decided to focus on the autumn only when considering the whole BS." Make clear to which extent this is a limitation of your analysis.

**We added "For studies needing the whole Barents Sea climatology in other seasons, other data sources are necessary.". Besides, including only autumn data for sure leaves out some information regarding the winter cooling and mixing. However, for the purpose here, which is a comparison with other studies, such as Lind et al., 2018, using autumn only is appropriate as other studies found in the literature also rely mostly on autumn data only (e.g., Skagseth et al., 2020, and I think also Lind et al., 2018).**

l107: PSU there is not "PSU" and salinity has no units, e.g. absolute salinity has g/kg, but even "regular" salinity is unitless

**We removed "PSU" in the manuscript.**

l110: "here not taken into account" why not? Apparently a factor up to two is involved. Is that a problem? Having a converging lat-lon grid for such a small area is questionable to begin with. Why this choice?

**We now take into account the narrowing of the longitudinal bands, thanks to the weighting function described in the manuscript. We also added this text: "Other atlas products, such as the WOA, are also provided on regular lat-lon grids, as well as most operational ocean models. Hence, it makes some of the usages more straightforward."**

ll113: "due to the cancelling effects from the increasing haline contraction and thermal expansion on density" -> due to the cancelling effects of increasing haline contraction and thermal expansion on density

**Done.**

l116: "at most" ???, the most?

**We now use "as much as possible" for clarity.**

page 7 Caption Fig6: "Average relative error on the Barents Sea for temperature"
-> "Average relative temperature error in the Barents Sea" (and similar for salinity)

**We changed both captions as "Average relative error on temperature/salinity in the Barents Sea".**

fig6 caption: "seasons" it is a function of time, not of seasons (the labels are years)

**We amended the caption as "as a function of time for each season".**

page 8 Fig7a caption "Average volumetric T-S diagrams during 1994–1998 and 2006–2010" is unclear, rephrase (the version in the text is clear)

**We changed the caption as: "Average of the volumetric T-S diagrams during both 1994--1998 and 2006--2010 periods"**

l118: "it is clear the error" insert "that"

**Done.**

l119: "This strengthens the reliability of the observed T-S changes." This is not clear to me, large uncertainties mean few data points, changes cannot be detected with few data points, so many changes may have gone unnoticed?

**The continuous fields from DIVA do not have fewer data points in case of large uncertainties, and a main point here is that the areas that see the largest changes also comprise the largest volumes of water which also have a good data coverage. Hence, large, unnoticed changes in the fringe areas would not have big impacts on the overall characteristics anyway (although it could of course prove important to local processes in those areas).**

page 9 l128: we focus on the periods

**Done.**

l131: "One way of studying changes in temperature and salinity in the MRA is to look at the vertical dimension."
I would remove this sentence. No additional information and the phrasing is not very "scientific" (e.g. you can "look at a piece of art", or "look at me, when I am talking to you", but I would study/inspect/analyse/take into account the vertical dimension).

**We removed the mentioned sentence.**

l131: "Temporal . . ." The temporal evolution . . . is shown . . .

**Done.**

Fig. 9, 10, 11: consider a different presentation of the data, e.g. a Hovmøller-like plot as in Fig 12 and 13 (except depth on the y-axis), the current plots are difficult to read. Maybe you can find a good way of combining Fig 9 to 13 in two or three panels. Now they take up a lot of space for limited information.

**We understand your suggestion but we finally decided to keep these figures unchanged, as we still think they provide a clear example of usage of the atlas, with a different approach with respect to the rest of the paper.**

page 9 l134: raise -> increase

**Done.**

l134: here and elsewhere: I am not a friend of abbreviations and I would consider spelling out Barents Sea every time you use "BS".

**Done.**

page 10 Section 6.3 It is not clear that the volumetric changes in T/S and density provide new information over the profiles (it get's warmer, salinity is ambiguous and density doesn't change very much), so the use of this section is not clear (and this has implications for the title of the paper, so I would ask for a better explanation of the volumetric t-s diagrams, etc.)

**We are providing examples of usages - in this case the calculation of water mass volumes enabled by the regular grid. Hence, the purpose is not to provide new information over (former) Section 6.2, but to demonstrate a different approach.**

eq(1) can only be a Ocean Heat Content (OHC) change, because deltaT is the change of temperature relative to a reference.

**We agree and changed the sentence accordingly.**

page 13 l150, 153: SI units are not supposed to be in italics

**Done.**

l153: "significant to the 95% level" not sure if this is the appropriate formulation

**We changed the sentence to: "significant at a confidence level of 95%".**

eq(2) [and to some extend eq(1)] what is delta s : s_ref-s? if so, then delta s/ (s+delta s) = s_ref-s/s_ref?

**Delta s is s-s_ref. We have clarified the text in this fashion.**

It is not clear what this EFWC is supposed to be. The proper (e.g. TEOS10) definition of freshwater is ocean-water minus salt (i.e. 1-s). In this sense, eq(2) can only be some fractional freshwater content (and just because it has been called freshwater before doesn't make it right). Because eq(2) depends on a reference salinity (the value of which is not even provided here), it is impossible to related the calculated numbers to anything else. Also the choice of reference (be it the mean as in your case or some arbitrary value) makes a difference in the time series. See Schauer and Losch (2019), their Figs3+4 for a simple illustration, also the discussion in Treguier et al (2014)

**Our objective is to illustrate that the atlas can be used to visualize the change in salinity or freshwater content. And even this way of calculating the freshwater content shows the temporal evolution (compared to the average freshwater content). For a**

**thorough investigation of the Barents Sea salt budget we agree that the suggested approach would be more appropriate. We have changed "EFWC" into "Ocean Freshwater anomaly" for greater clarity. Besides, the reference salinity is now provided in the manuscript to facilitate comparisons.**

Similarly the OHC in eq(1) depends on the reference (and the units, do you use degC or Kelvin?). In the OHC case one can argue that everyone in oceanography uses degC and a reference of 0degC to compute OHC so that the ambiguity problem goes away (see McDougall, 2003, doi:10.1175/1520-0485(2003)033<0945:PEACOV>2.0.CO;2). Here the reference appears to be the mean temperature resurrecting the same problem as for the salinity anomaly/fractional freshwater.

**We now provide the reference temperature in the manuscript to facilitate comparisons. Delta t is a difference of two temperatures in Celsius degrees.**

161: SI units not in italics

**Done.**

l163: "For both OHC and EFWC trends significance, we followed the Fisher–Snedecor test described in Chouquet (2009) and Montgomery et al. (2012) augmented by a penalization of autocorrelation (Wilks, 1995)"
this information should have come earlier, also: "For both OHC and EFWC the significance of the trends was determined following . . ."

**We modified the manuscript accordingly.**

page 14 l165: any idea or comment why the salinity trend at BSO is opposite to "EFWC"? Maybe because of the minus sign in the definition? Wouldn't it make more sense to reverse the sign in the plot to illustrate the correlation?

**The freshwater content is negatively correlated to the salinity, and we agree this relies in its definition. Although we agree reverting the axis in the plot is a possibility, we chose to keep it unchanged to be consistent with the negative correlation.**

This also goes back to my point of eq(2): Using salt content (integral over salinity) would be a less ambiguous measure and would yield itself much more easily to physical interpretation.

**See the previous answers on this topic.**

page 14 Conclusions
the conclusions are weak, but since this is a data product, there may not have to be strong conclusions about the physical interpretation. I would focus on the presentation of the data in the conclusion.

**We agree and added some text focusing more on the data product: "These error fields can be used to exclude unreliable periods of areas, as shown by the examples of**

**usage provided in this study. Besides, the regular grid facilitates the computation and the visualization of various metrics such as profiles, volumetric T-S diagrams or OHC and OFW.”**

l169: on this part -> in this part

**Done.**

l169: “much” replace by “some” or remove

**Done.**

l170: “provided a variational method minimising the expected errors on the resulting fields is used” I don't think that this research shows that this method is required for the analysis. To be able to draw this conclusion I would like to see why it is impossible to extract physical information from sparse data without this interpolation method. Please rephrase.

**We rephrased as follows: “Although the in situ data is sometimes scarce in this part of the Arctic, we show here that physical information can still be extracted from compiled databases by using a variational method minimising the expected errors on the resulting fields.”**

Please also note the supplement to this comment:
https://www.earth-syst-sci-data-discuss.net/essd-2020-70/essd-2020-70-RC2-supplement.pdf

**Reply to Interactive comment on "A volumetric census of the Barents Sea in a changing climate" by Sylvain Watelet et al.**

We would like to thank the Referee #3 for the time spent in the review of our work. Hereafter, we answer **in bold** to the comments.

Comment and reply:

Anonymous Referee #3

This paper presents a novel temperature and salinity datasets that is presented on a regular grid for the Barents Sea, which are "constructed" from the available datasets. It also presents basic analysis of the thermohaline trends that are showing the state of the Barents Sea. The manuscript is well written and is easy to follow. For these reasons, I think that presented manuscript has a potential for the publication in the ESSD.

**We would like to thank the Referee #3 for this positive assessment of our work.**

However, I have some concerns regarding the manuscript, especially when it comes to the description of the used methods to "construct" the maps of temperature and salinity. In the manuscript, the Authors mention the DIVA software package, however I am missing the details on the used methods and the existing data. My other main concern is the coverage (in space and time) of the used thermohaline data, as well as for the final gridded maps, that were not clearly described in the manuscript, and for which I think that is important for the reader to understand your choice of analysis that are limited to seasons, years and/or regions. Altogether, I recommend this manuscript for the publication after the major revisions.

The main concerns from my side are:

1. In the manuscript, I find that Section 2 is poorly described, which could lead the readers to miss the important details of the used data. You could provide a Figure of the Barents Sea with the locations of the used in situ data, and others. Are these data measured regularly (for instance on a monthly basis)? In addition, the Authors could also provide the references for the WOD13 data, PINRO CTD data and the NPI data.

**As the reviewer suggests, we have provided more extensive information about the data sources, including some information on the regularity of the data gathering and reasons for variable data coverage between years:**
**The data coverage is usually better in the spring (Feb-Mar-Apr) and autumn (Aug-Sep-Oct) seasons compared with the rest of the year due to extensive survey activity during these seasons. However, while the surveys generally cover the ice-free area of the Barents Sea, the spatial coverage vary between years and the coverage is usually more extensive in the autumn compared with the spring. Moreover, while data from the annual spring and autumn surveys in the Barents Sea are obtained on a regular grid, data from other surveys are more focused in smaller areas or along fixed**

**sections.**

2. The description of the method could include more details. Aside from the DIVA package, you mentioned several techniques that you used to create the gridded data with the estimated error fields, however, I am missing the details of the methods. For instance, until now I have never heard of the "clever poor man's method", and I couldn't get access to a reference paper. You could provide additional information about the used methods, since this is the main work of your manuscript.

**We added this text on the DIVA method: "In practice, the aim of the VIM is to minimize the following cost function J:**
**[equation, see manuscript]**
**where the N d observations d j are used to reconstruct the analysed field φ and with [equation, see manuscript]**
**where α0 penalizes the field itself (anomalies with respect to a reference field, e.g., a climatological average), α1 penalizes gradients (no trends), α2 penalizes variability (regularization), and μ j penalizes data-analysis misfits (objective analysis) (Troupin et al., 2016). Unless specified otherwise, we always use the command line version of DIVA in this study. This version comes with the full set of options, for instance regarding the optimization of the statistical parameters later used in the analyses.". Regarding the clever poor man's error, we added this explanation: "The poor man's error is computed by analysing a "data" vector with unit values and is very cost-effective (Troupin et al., 2010), but the error field is too optimistic. It is shown that using the same method with a correlation length divided by a factor ~1.7 requires a similar computation time and yields a more realistic estimate of the error, that is, the clever poor man's error."**

3. Also, just by reading the Sections 3, I find it a bit confusing, which is a bit disappointing, considering that this Section is containing the main core of your paper. First of all, you should make it clear for the readers what are the input data and what are the resulting gridded maps. Similar to my previous comment, I am missing the description of the datasets. What is the coverage of the original datasets, and what is for the resulting gridded maps (in space and time)? Are the resulting maps on a monthly basis? In L57: you mention that "the ODV spreadsheets were vertically interpolated onto 23 depths. . .", however, I am missing the information on how the spreadsheets were constructed. Do they contain only the original data, and if so, what does one spreadsheet represent and how many data does it contain? Also, you mentioned that you performed the analysis on four seasons (L67), and that you constructed the data for these analysis by using an 11-year windows (for which I am not sure whether you averaged only the corresponding seasons during these 11 years, or the whole year). At this point, I was not sure whether this analysis is performed on the data that are showed in the T-S diagrams? I have to admit, I was bit confused reading this Section, so my suggestion is to rewrite it in a form that is more precise, and providing more details on the used methods, including the methods used for estimation of the errors.

**We followed the suggestion of the Referee #3 and rewrote the Section 3. The Section**

**now includes more details on the variational inverse method, as well as on the method to estimate the errors. The Section 2 was also rewritten and now includes more details on the data sources. Regarding the data coverage, the Section 4 now includes further information on the data gaps in the atlas. The resulting atlas is provided on a seasonal basis (i.e. 3 months), as defined in Section 3. The ODV spreadsheets were simply exported from the software ODV after importing the original datasets in the software, which is a convenient way of making many datasets readable by DIVA. These spreadsheets only contain original hydrographic data that are now described more extensively in Section 2. We now make more clear in the text that each seasonal analysis relies on a 11-year reference field that corresponds to the same season only. The T-S diagrams are based on the resulting maps, that is the temperature and salinity atlas. We clarified the text in Section 5 to explain that these are examples of usage of the atlas.**

4. Regarding the basic analysis of the gridded thermohaline fields, I find these Sections too descriptive. You should provide numbers when making statements, i.e. whenever you use phrases such as "increase", "decrease", "trend", etc., you should a give a value by "how much". Some examples are given bellow in the detailed comments

**We agree with the "how much" issue, and we now quantify the changes wherever possible throughout the manuscript. We have also written an introduction to Section 5 stating that the analysis presented provide examples of usage of the data product and its features, rather than an in-depth analysis of the climatic conditions. The text reads: "*In the following sections we demonstrate how the error field provided in the atlas can be utilized to objectively limit the data in time or space before applying the desired analysis. Moreover, we give some examples of possible usages of the atlas product.*"**

Minor detailed comments and suggestions:

Figures:

1. Most of the Figures are lacking the descriptions. Each of the Figure should contain detailed description of what it is showing. For instance, in Fig.2 and 3 it is not clear for me what the time-step for the data is. On Fig. 4 and 5, you should provide explanation for the values on the colorbar. On Fig. 7 should be mentioned which type of data is shown.

**On Fig. 2 and 3, we added "(seasons)" for greater clarity. We added the following explanation to Fig. 4 and 5: "This variable measures the added value brought by in situ data to the analysis: 0 would be the true field while 1 corresponds to an absence of data, that is an analysis equal to the first guess.". For Fig. 7, we changed a part of the caption as "Average of the volumetric T-S diagrams during both 1994–1998 and 2006–2010 periods.".**

2. In Fig6a the y-axis: "Averaged relative temperature error" or "Averaged relative error for temperature". Same for the salinity (Fig6b)

L15 and elsewhere: I don't understand the usage of BS acronym. However, if you chose it, you should be more punctilious while using it. In this line you mention "Barents Sea", without the acronym, and you introduce it in the L16. Throughout the manuscript you are sometimes using "Barents Sea", and sometimes "BS". This should be corrected also at the Figure 1. Same comment applies for all the other used acronyms.

**We now use "Barents Sea" everywhere in the manuscript.**

L17: use the apostrophes when mentioning Atlantification. As far as I know, this is not a name for a physical process, even though the readers do understand the meaning of the phrase.

**Done.**

L18: what are the "both physical conditions"? Perhaps you can exclude "such as", since it implies that there are more than two

**We changed the text to "its physical conditions".**

L18 and 19: as well as on biological and marine ecosystem

**Done.**

L30: varies between seasons and years, especially during winter and spring

**Done.**

L31: . . . or concentrated at fixed sections.

**Done.**

L32: . . . sea surface temperature, and recently sea surface salinity

**Done.**

L33: Don't use "E.g." at the beginning of the sentence. Instead you could say "For example,. . ."

**Done.**

L34: . . . the Arctic that shows temperature increase for the period... What was the increase?

**We changed the text accordingly. Comiso and Hall (2014) do not provide a precise figure for the northern Barents Sea, but their Fig. 2a shows the spatial distribution of the increase.**

L35: . . . between the two periods: 1979-1995 and 1996-2012.

**Done.**

L36: . . .property changes,

**Done.**

L36: I find the phrase "in situ" differently written at several places. Sometimes it is "in situ", and sometimes is "in-situ". Try to be consistent when using phrases. I found in many previous papers written in curves "in situ"

**We now use this last suggestion throughout the manuscript.**

L36 and 37: . . . often have disadvantages of. . .. and/or time (sometimes it could be both)

**Done.**

L37 and 38: Please rephrase the sentence, it doesn't sound grammatically correct.

**We changed the sentence as: "Thus, providing these observations on a regular grid is desirable in order to examine spatio-temporal changes."**

L39 and 40: Does the seasonal temporal resolution mean a 4-month averages? You should be more concise here, as well as in Section 3 (See my comment 3 in the main concerns)

**These are not averages but analyses made per season using at once all in situ data corresponding to each particular season, that is, 3- month periods.**

L40: "based on all available observations". Does this also include satellite data? It is not that clear in the Section 2.

**We only used in situ data, this is now clarified in the sentence.**

L53: . . . in situ data using a Variational. . .

**Done.**

L63 to 65: To which data are you referring here, original or the ones already inter-polated on the 23 layers? It is not easy to follow this Section. At this point I am understanding that the Fig 2 and 3 are showing the interpolated data. However, L104 suggests that these Figures show original measurements. I find it a bit confusing.

**We now use "All the interpolated data" and "the amount of exploitable data" to make it clearer that we refer to vertically interpolated and cleaned data.**

L67: You should define the periods for the 4-month averages, just to be more precise. Later on, you use "autumn", and you never defined that season. Even though it is self-clear, I find it better to be as much as precise as possible when writing a scientific paper. My suggestion "November to January (autumn), etc."

**Done.**

L96: from the mid-2000s. Also, to what period do you refer when saying "than previously"?

**Following a comment from another reviewer, we changed this sentence into "Lind et al. (2018) provided some evidence suggesting a warmer and saltier northern Barents Sea since the mid–2000s."**

L97: Why is the reason of choosing these exact 5-years periods? Also, could you give an exact number of the data used in the analysis?
**The rationale is provided in this added sentence:**
*We limit our analysis to comparing the two 5-year periods between 1994-1998 and 2006-2010, where the former represents a relatively cold period while the latter represents a warm period relative to the last 50 years.*

L100: Why didn't you include year 2016 for the estimation of the error fields?

**The atlas is available on 4 seasons between 1965 and 2016, which are defined as follows: November to January for winter, February to April for spring, May to July for summer, and August to October for autumn. The first season is thus November 1964 to January 1965, the last being August to October 2016. Considering this, we preferred to make the average between spring 1965 and winter 2015-2016, in order to have only full years running from February to January. Otherwise, there would be two biases involved: no data in November and December 1964 and no data for November and December 2016. For greater clarity, we included this reasoning in the manuscript.**

L102 and 103: The relative error field averaged through all the layers for each variable and season is shown in Fig. 6

**Done, with a small change to fit a comment of another reviewer: "The relative error field averaged over all layers for each variable and season is shown in Fig..."**

L105: there is no need to say "only when considering the whole BS", since this is the only analysis that is considering the whole BS.

**Maybe there is a misunderstanding here, what we mean is that we only used data from the autumn for our whole Barents Sea analysis.**

L106: Volumetric T-S changes for both periods were carried out by summing all the pixels falling inside the T-S classes . . . having a step of . . .

**We partly implemented your suggestion as "Volumetric T-S diagrams for both 1994--1998 and 2006--2010 were carried out by summing all the pixels falling inside the T-S classes defined by temperature ranging from -1 to 7 °C and salinity varying between 33 and 35.5 having a step of 0.05 °C and 0.025, respectively." since we first computed T-S diagrams for each period, the changes are only computed afterwards.**

L106: Does the data only correspond to the autumn data? You should also mention this in the description of the Fig. 7

**Indeed, the analysis for the whole Barents Sea only uses autumn data. We added "For all panels, only autumn is used." in the Fig. 7. It is also mention in the following, added sentence:** *"[...] the data coverage is generally better and, hence, the error is generally smaller in the autumn season compared with the other seasons. [...] For this reason, we decided to focus on the autumn only when considering the whole Barents Sea."*

L112 to 113: You should mention that this increase implies only to autumn. Also, this is a good example of descriptive sentence. You should provide averages: "the increase in temperature and salinity is clear, by XX C and XX PSU in average"

**We now mention the autumn and added the following sentence: "Between the T or S classes showing the highest change, there is temperature shift of 5° C and a salinity shift of 0.2."**

L115: In Section 3 u stated that the reference fields were defined by an 11-year window, and here you say that you used a 10-year window. Please provide additional explanation.

**This was a typo, we corrected it.**

L117 to 119: . . . weighted by the layer. . . for periods 1994-1998 and 2006-2010. It is clear that the error is much lower on the T-S classes showing larger changes, which are . . . This strengthens the reliability of the observed autumn T-S changes in the BS.

**We revisited the text as follows: "Further utilizing the error field, we provide an estimation of the uncertainties for both the two 5-year periods included in the above analysis. Comparing the error fields in both periods (Fig. 5c, d) with the changes in the T-S properties between the two periods (Fig. 5b), as well as the T-S diagrams of both periods (Fig. 5a), it is clear that the error is small for the T-S classes that have the largest presence and also are showing the largest changes. This strengthens the reliability of the findings of T-S changes in the Barents Sea in autumn."**

L122: Rephrase the sentence and correct the grammar

**We corrected the sentence a follows: "The MRA encompasses the southern part of the Barents Sea which is dominated by the Atlantic Water inflow and kept ice-free year round, hence the better data coverage in all seasons."**

L125 to 127: If the errors are not shown, you need to state so

**The errors are shown, they are displayed in the Supplementary Material.**

L127. This advantage allows us to analyze all the seasons at the MRA, in contrast to the whole BS. Here we focus on the periods: 1965-2015 for the temperature and 1970-2010 for the salinity. Also, add a sentence as an explanation of using these periods.

**In order to also answer to comments from other reviewers, the text is now the following: "The MRA encompasses the southern part of the Barents Sea which is dominated by the Atlantic Water inflow and kept ice-free year round, hence the better data coverage in all seasons. This allows us to analyze all the seasons in the MRA, in contrast to only the autumn season when analyzing the whole Barents Sea (see section 5.1), with the exception that for salinity the data coverage is sufficient only for the period 1970—2010. For temperature, we use the period 1965–2015. In addition, there are gaps in the salinity data during the 1971—1972 winter period and in both temperature and salinity data during the 1996–1997 winter period."**

L128: You state that the years 1996-1997 is having a lack of data for the temperature at the MRA, which is showing the lowest errors. However, those two years are within the 5-years period that you used in the analysis of the previous Section, where you showed an increase in temperature. At this point, I am not convinced in the reliability of the previous analysis, and even more I still don't understand the choice of those 5-year periods for that analysis. Could you please explain?

**Considering 1996-1997, we only refer to the winter which has a lack of data. The previous analysis on the whole Barents Sea only uses autumn data. Since there are much more data in autumn than in winter, this is consistent. Regarding the choice of years, please see the answer provided above.**

Subsection 6.2: Why did you choose to show vertical seasonal thermohaline profiles, all in one plot? The figures are a bit "messy", showing all seasons, and it is not that clear to depict the trends. Instead of the profiles, you could show surface plots with the estimated trends for three different averaged layers (0-50, 50-300, and 300 to bottom). Other choice could be (a), (b), (c), and (d) for the profiles, where you could separate 4 seasons.

**We want to show profiles, as the stratification in the Barents Sea is an important parameter, which we want to display along with the changes within each depth layer. We understand the suggestion to split the profiles into seasons but eventually decided to keep the figures unchanged, because despite the inter-seasonal variability, the larger time scale changes can be seen thanks to the color scale made on a yearly basis.**

L131: I don't understand the point of this sentence. You should remove it.

**Done.**

L132 to 134: I also find this descriptive. What is the averaged values for the temperature increase? Also for the salinity, "unambiguous raise between the 90s and the 2000s", how much? . . . "similarly to the observation made for the whole BS", I am missing a reference here.

**We now quantify these changes in the following sentences: "The temperature gradually increased throughout the whole water column during the period 1965–2015, by 1.74° C on average. For salinity, matters are not so clear, except the unambiguous average increase of 0.11 between 1990–2010, similarly to the observation made for the whole Barents Sea between the 1990s and the 2000s." Regarding "similarly to the observation made for the whole BS", we refer to the observed increase of salinity in the Barents Sea between the two 5-year periods in the 90s and 2000s (autumn only).**

Subsection 6.3.: Provide additional information on how you estimated the volume of water? What exactly are the Figures 12, 13 and 14 showing? Once again, I find this paragraph too descriptive. When using phrases such as "increase", "decrease", "trend", etc. you should a give a value by "how much".

**We added this additional information on the estimation of the water volume: "The calculations follow a method similar to Section 5.1". The following sentence was also added: "The aim is to show the relative volume occupied by each temperature and salinity class." Further, we now include "per season" for each of these figures. Finally, we now quantify the changes as follows: "For instance, between the periods 1975–1985 and 2005–2015, the relative volume occupied by temperature below 0° C decreased from 19.64% to 1.77%."; "For instance, between the periods 1975–1985 and 2000–2010, the relative volume occupied by salinity below 35 decreased from 86.84% to 62.67%."; "However, water masses with densities above 1028.0 kg m −3 , associated with dense water production, has rarely exceeded 20 percent of the total water mass within the MRA after year 2000.".**

L139: I don't understand the phrase "classes" in this sentence. Define the "classes".

**We have added the step of each class in order to define them more explicitly.**

L139: The calculations. . .

**Done.**

L145: . . . similarly to the conclusions made in Section 6.2.

**Done.**

Fig 12, 13 and 14: Are the diagrams showing the sums of the volumes in a whole seasons? You should state that in the description of the Figures

**Yes, we added "per season" for each of these figures.**

L146: . . . at the MRA. . .

**Done.**

L151: Define "reference period"

**Done (1970-2010).**

L152: In the Formula (1), OHC is dependent on the density changes, which is dependent on both temperature and salinity changes. How could you estimate OHC value for the period outside 1970-2010? Even in the L149 you stated that t and s are between 1970-2010. Could you please explain?

**For the OHC, we used an averaged density per grid point over 1970-2010. This is now stated in the text as: "ρ is the density of seawater averaged over 1970–2010 for each grid point".**

L154: Is the correlation significant? From the Figure 15a, it clearly is, but it is better to add it in the sentence as well.

**Yes, we added this information in the text.**

L155: . . . at the BSO

**Done.**

L161: In Fig. 15b changes in the EFWC . . .

**Done.**

L163 and 164: To which threshold do you refer? The sentence on the choice of the significance tests should be stated before.

**We mean the significance threshold. We clarified this and detailed the significance test earlier in the manuscript.**

L165: Is the correlation significant? If yes, could you give a sentence in explanation on why the correlation is negative, similar to the one you gave for the OHC and temperature positive correlation? Also, it would be interesting to know what caused an extreme salinity decrease during the late 1970s and early 1980s, evident in both EFWC increase (Fig15b) and in the percentage of the total volume (Fig13). Are there any references for that?

**Although very close to the significance level, the correlation is not significant. We have provided a reference to the propagation of the so-called "Great Salinity Anomaly" through the Barents Sea in the late 1970s and 1980s: "[...] the low salinity**

**associated with the "Great Salinity Anomaly of the 1980s" (Dickson et al., 1988) is seen as a distinct maximum of salinities below 34.8."**

L170: Rephrase the sentence

**We rephrased as follows: "Although the in situ data is sometimes scarce in this part of the Arctic, we show here that physical information can still be extracted from compiled databases by using a variational method minimising the expected errors on the resulting fields."**

L171 and 172: "The results are consistent with the recent "Atlantification" processes at the BS already observed in the previous studies, i.e. warmer and more saline BS, even though our analysis only includes autumn when considering the whole BS". Also, I am missing a references for the previous studies

**Done, we also added two references for the previous studies.**

L172 to 175: Concentrating on the MRA in the BS allowed us to analyze longer period (1965-2015) with all seasons included. The analyses showed similar results to the ones made for the whole BS, showing an overall positive temperature and salinity trend (with numbers!), while . . . cancelling effects of both temperature and salinity increase.

**We adapted the text as: "Concentrating on the MRA in the Barents Sea allowed us to analyze longer period (1965–2015) with all seasons included. The analyses showed similar results to the ones made for the whole Barents Sea, showing an overall positive temperature and salinity trend, that is +1.74° C between 1965–2015 and a salinity increase of 0.11 between 1990–2010."**

L176 and 177: . . .these conclusions as they show positive and negative trend, respectively, during the period 1965-2016. I am a bit concerned here. As I stated before in my comments, I find this period suspicious. Moreover, EFWC was estimated only for the period 1970-2010. In addition, I don't think that EFWC trend is significant, since R2 is only around 12%.

**We agree and changed the text as follows: "Finally, the computation of OHC and OFW are consistent with these conclusions as they show positive and negative trend, respectively, during the period 1965–2015 for the OHC and 1970–2010 for the OFW, although the latter trend is not significant."**

Please also note the supplement to this comment:
https://www.earth-syst-sci-data-discuss.net/essd-2020-70/essd-2020-70-RC3-supplement.pdf

[revised manuscript text omitted]

---

## Author Response (AR2)

**Thank you for noticing the mistake, our answer is in bold.**

A very small check:
line 207: Mork et al. (2019) - should be - (Mork et al. 2019).

**We changed the manuscript accordingly.**

[revised manuscript text omitted]